# The microtubule polymerase Stu2 promotes oligomerization of the γ-TuSC for cytoplasmic microtubule nucleation

**Judith Gunzelmann, Diana Rüthnick, Tien-chen Lin[†], Wanlu Zhang, Annett Neuner, Ursula Jäkle, Elmar Schiebel\***

Zentrum für Molekulare Biologie der Universität Heidelberg, DKFZ-ZMBH Allianz, Heidelberg, Germany

**Abstract** Stu2/XMAP215/ZYG-9/Dis1/Alp14/Msps/ch-TOG family members in association with with γ-tubulin complexes nucleate microtubules, but we know little about the interplay of these nucleation factors. Here, we show that the budding yeast Stu2 in complex with the γ-tubulin receptor Spc72 nucleates microtubules in vitro without the small γ-tubulin complex (γ-TuSC). Upon γ-TuSC addition, Stu2 facilitates Spc72–γ-TuSC interaction by binding to Spc72 and γ-TuSC. Stu2 together with Spc72–γ-TuSC increases microtubule nucleation in a process that is dependent on the TOG domains of Stu2. Importantly, these activities are also important for microtubule nucleation in vivo. Stu2 stabilizes Spc72–γ-TuSC at the minus end of cytoplasmic microtubules (cMTs) and an in vivo assay indicates that cMT nucleation requires the TOG domains of Stu2. Upon γ-tubulin depletion, we observed efficient cMT nucleation away from the spindle pole body (SPB), which was dependent on Stu2. Thus, γ-TuSC restricts cMT assembly to the SPB whereas Stu2 nucleates cMTs together with γ-TuSC and stabilizes γ-TuSC at the cMT minus end.
DOI: https://doi.org/10.7554/eLife.39932.001

**\*For correspondence:**
e.schiebel@zmbh.uni-heidelberg.
de

**Present address:** [†]Deutsches Zentrum für Neurodegenerative Erkrankungen (DZNE), Bonn, Germany

**Competing interests:** The authors declare that no competing interests exist.

## Introduction

Microtubules (MTs) are cylindrical polymers composed of tubulin, a heterodimer of α- and β-tubulin, with β-tubulin facing the more dynamic MT plus end and α-tubulin the less dynamic MT minus end (*Mitchison, 1993*). One factor that promotes MT nucleation and that resides at the MT minus end is the γ-tubulin complex, which is composed of γ-tubulin and additional γ-tubulin complex proteins (GCPs) (*Zheng et al., 1995*). The γ-tubulin small complex (γ-TuSC) of budding yeast contains γ-tubulin Tub4, GCP2/Spc97 and GCP3/Spc98. The γ-TuSC oligomerizes into a left-handed spiral upon interaction with the γ-tubulin receptor protein Spc110 at the nuclear side of the yeast spindle pole body (SPB) (*Geissler et al., 1996*; *Knop and Schiebel, 1997, 1998*; *Kollman et al., 2010*; *Nguyen et al., 1998*). In human cells, the γ-TuSC, together with GCP4, GCP5 and GCP6, forms a stable γ-tubulin ring complex (γ-TuRC) (*Zheng et al., 1995*). Through the interaction of γ-tubulin with α/β-tubulin, the kinetic barrier of α/β-tubulin oligomerization is overcome and MT assembly is facilitated (*Kollman et al., 2011*; *Roostalu and Surrey, 2017*).

Stu2, as well as XMAP215/ZYG-9/Dis1/Alp14/Msps/ch, belongs to a family of proteins that contain TOG (Tumour Overexpressed Gene) domains (here collectively called TOG-domain proteins) and function as MT polymerases. They localize to both the plus and the minus end of MTs (*Ayaz et al., 2012*; *Ayaz et al., 2014*; *Brouhard et al., 2008*; *Podolski et al., 2014*). The two TOG domains at the N-terminus of budding yeast Stu2 can directly bind α/β-tubulin. Further on, Stu2 can be recruited to MTs via its C-terminal MT-binding site. This interaction of Stu2 with MTs recruits the α/β-tubulin heterodimer to the growing MT plus end, promoting MT polymerisation (*Al-Bassam et al., 2006*; *Ayaz et al., 2012*; *Ayaz et al., 2014*; *Brouhard et al., 2008*).

Proteins such as the human 'Transforming Acidic Coiled Coil' (TACC) protein CDK5RAP2, the *Caenorhabditis elegans* SPD-5 or the budding yeast γ-TuSC receptor protein Spc72 target TOG domain proteins to sites of MT nucleation at the centrosome and the SPB (*Chavali et al., 2016*; *Chen et al., 1998*; *Cuschieri et al., 2006*; *Raff, 2002*; *Sato et al., 2004*; *Usui et al., 2003*; *Woodruff et al., 2017*). In vitro studies indicated a role for TOG domain proteins in MT nucleation, in cooperation with either the γ-tubulin complex or the MT binding protein TPX2 (*Roostalu et al., 2015*; *Thawani et al., 2018*; *Wieczorek et al., 2015*). Just recently, it was suggested that XMAP215 has the ability to interact with purified human γ-tubulin via its C-terminal MT-binding domain (*Thawani et al., 2018*). Furthermore, an in vitro study suggests that the *C. elegans* SPD-5 forms a gel-like matrix, which is able to recruit ZYG-9 and the microtubule stabilizing protein TPXL-1 (TPX2 homolog). These proteins together concentrate α/β-tubulin, leading to MT nucleation in form of asters, without the action of γ-tubulin (*Woodruff et al., 2017*). The emerging picture is that TOG-domain proteins have the ability to nucleate MTs in vitro. However, the following open questions remain: (1) Do TOG domain proteins have the ability to nucleate MTs in vivo independently of γ-tubulin complexes? (2) Do TOG domain proteins have structural functions at the MT minus end? (3) Is the role of TOG-domain proteins in MT nucleation context specific (*Roostalu and Surrey, 2017*)?

The *Saccharomyces cerevisiae* SPB provides an ideal system to address these questions because it has spatially distinct MT nucleation sites at the SPB. The SPB is embedded in the nuclear envelope and only the cytoplasmic side recruits Stu2 through binding to the γ-TuSC receptor protein Spc72 (*Chen et al., 1998*). By contrast, the pericentrin orthologue Spc110 localizes and oligomerizes γ-TuSC on the nuclear side of the SPB without Stu2 (*Kilmartin and Goh, 1996*; *Knop and Schiebel, 1998*; *Lin et al., 2014*; *Sundberg et al., 1996*). Spc110 has a well-established Centrosomin Motif 1 (CM1) that interacts with γ-TuSC. Interestingly, the cytoplasmic Spc72 has only a degenerated CM1 whose function has not been investigated (*Chen et al., 1998*; *Usui et al., 2003*).

Here, we analysed the role of Stu2 at MT nucleation sites and its cooperation with γ-TuSC. Unlike the Spc110–γ-TuSC interaction, we found that γ-TuSC binding to Spc72 is of relatively low affinity. However, binding of Stu2 to Spc72 increases the affinity for γ-TuSC 20-fold. Consistent with these in vitro data, Stu2 stabilized the γ-TuSC cap at the minus end of single cytoplasmic microtubules (cMTs) in vivo. Furthermore, in vitro reconstitution of MT assembly indicated that the purified Stu2–Spc72 complex was already sufficient to nucleate MT asters. Addition of recombinant γ-TuSC to the Spc72–Stu2 complex promoted γ-TuSC oligomerization into a functional MT nucleation template independent of the TOG domains of Stu2. However, efficient MT nucleation from these Spc72–Stu2–γ-TuSC complexes required the action of the TOG domains. The reason could be that the TOG domains of Stu2 facilitate binding of the first α/β-tubulin subunits to the γ-tubulin template. Complementary in vivo studies supported the role of Stu2 in MT nucleation. The depletion of γ-tubulin Tub4 did not inhibit cytoplasmic MT (cMT) nucleation, but these MTs assembled in the cytoplasm away from the SPB and were strictly dependent on Stu2. Together, this study illustrates specific roles for Stu2 and γ-TuSC in MT nucleation and in the maintenance of the γ-TuSC oligomer at the cMT minus end.

## Results

### The CM1 of Spc72 mediates binding to γ-TuSC

The CM1 motif of Spc110 interacts with γ-TuSC and is essential for its oligomerization into an active nucleation template (*Lin et al., 2014*). Sequence alignments indicate that the N-terminus of Spc72 carries a CM1 motif (63–88 aa) that is, however, shorter (25 amino acids in Spc72 versus 78 in Spc110) than the canonical CM1 motifs in budding yeast Spc110, *Drosophila* CNN and human CDK5RAP2 (*Figure 1a*) (*Lin et al., 2014*; *Lin et al., 2015*; *Usui et al., 2003*). The CM1 of Spc72 probably only reflects the CM1a portion of Spc110 while CM1b is missing (*Lin et al., 2016*). We asked whether the truncated CM1 of Spc72 is essential for its function by mutating three conserved residues to alanine (spc72$^{CM1}$; K75A, K77A, E83A) (*Figure 1a*). spc72$^{CM1}$ did not support the growth of yeast cells, indicating the essential function of the CM1 motif (*Figure 1b*).

Amino acid substitutions in CM1 of *Candida albicans* Spc110 or human CDK5RAP2 impair binding to the γ-tubulin complexes (*Choi et al., 2010*; *Lin et al., 2016*). To test for the interaction of Spc72

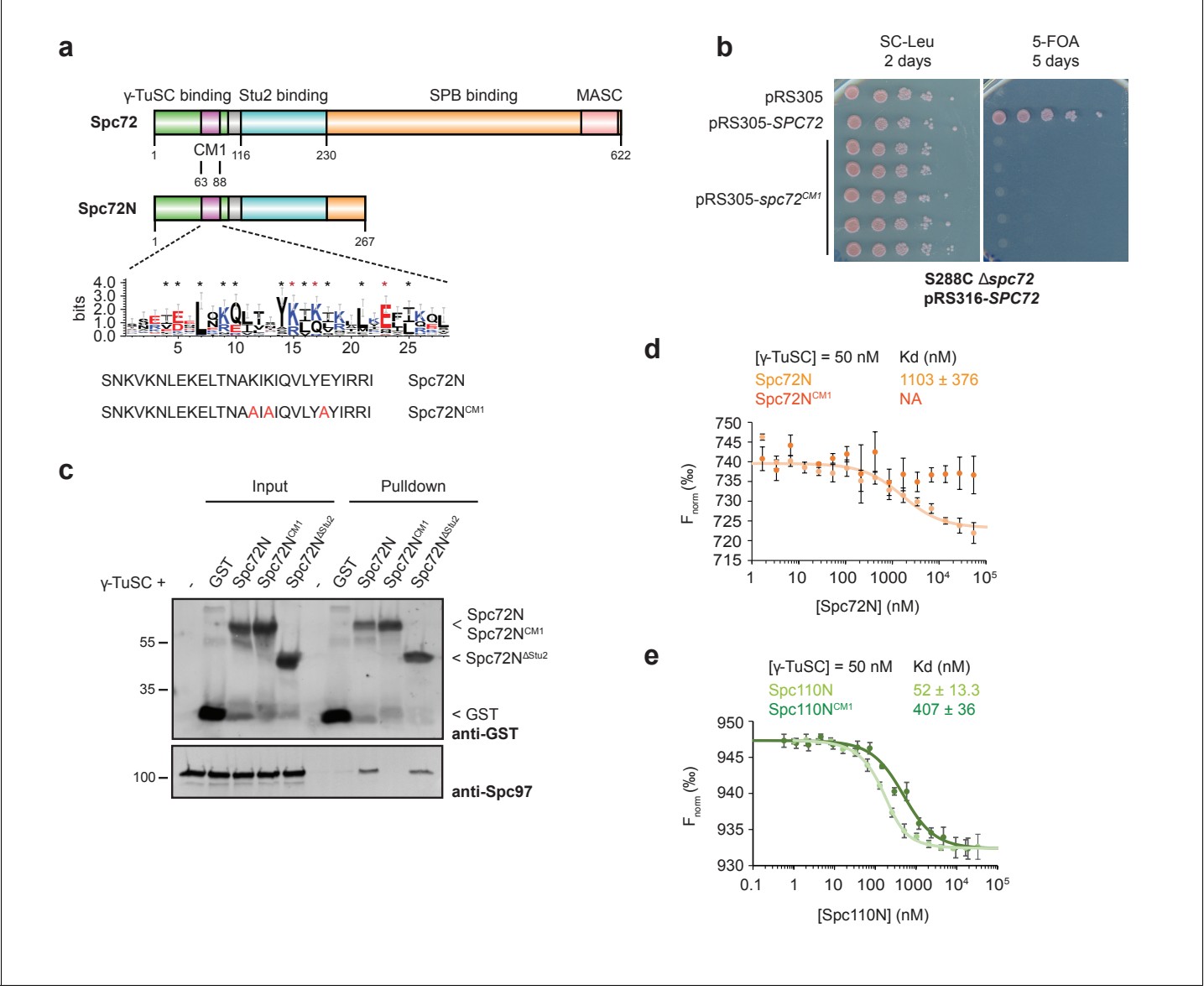

**Figure 1.** Binding of Spc72 to γ-TuSC. (a) Substructure of Spc72. An enlargement of the putative CM1-domain of Spc72 is shown (designed with WebLogo 3.5.0; taken from [*Lin et al., 2014*]). Mutations in Spc72N^CM1 are highlighted in red. Abbreviations: CM1, Centrosomin Motif 1; MASC, Mto1 and Spc72p C terminus. (b) Mutations in the CM1 of *SPC72* affect the viability of yeast cells. Cells were spotted as 10-fold serial dilutions. (c) In a pulldown experiment, purified Spc72N^CM1 failed to bind to γ-TuSC while GST-Spc72N^ΔStu2 showed binding. Immunoblots that were developed with anti-GST and Spc97 antibodies are shown. (d) Microscale thermophoresis (MST) measurements of Spc72N and Spc72N^CM1 binding to γ-TuSC. Mean ± SD, n = 6; NA, not applicable. (e) MST measurements of the binding of Spc110N and Spc110N^CM1 to γ-TuSC. Mean ± SD, n = 3.

DOI: https://doi.org/10.7554/eLife.39932.002

with the γ-TuSC,we designed constructs expressing the N-terminal portion from amino acids (aa) 1–267 (Spc72N), a mutant CM1 version (Spc72N^CM1; aa 1–267 with K75A, K77A, E83A) and a Stu2-binding deficient mutant (Spc72N^ΔStu2; Spc72N with deletion of aa 116–230). In pulldown experiments, γ-TuSC showed reduced binding to Spc72N^CM1 in comparison to Spc72N or Spc72N^ΔStu2 (*Figure 1c*, Spc97-blot). Quantification of the interaction affinities by microscale thermophoresis (MST) determined the dissociation constant ($K_d$) of the Spc72N–γ-TuSC interaction with the high value of 1.1 µM, which reflects a relatively low binding affinity (*Figure 1d*). The Spc72N^CM1 mutant protein did not show binding to γ-TuSC in this assay (*Figure 1d*). Importantly, the binding affinity of γ-TuSC to the truncated N-terminal construct of Spc110 (Spc110N, aa 1–220) was 20-fold higher

than that of Spc72N (compare *Figure 1d and e*). Also in case of Spc110, a mutant CM1 version (Spc110N<sup>CM1</sup>, K120Q; E121A) showed reduced binding affinity (*Figure 1e*). In conclusion, Spc72 can bind the γ-TuSC dependent on the CM1, but with a 20-fold lower affinity than Spc110.

## Stu2 binding to Spc72 enhances interaction with γ-TuSC

The relatively low affinity (high $K_d$) of Spc72 for γ-TuSC raises the question of how this receptor recruits the γ-TuSC to the SPB efficiently. Stu2, which interacts with Spc72, could mediate the binding of γ-TuSC to Spc72. To test this notion, we quantified the interactions between Spc72 and Stu2, as well as the Spc72–Stu2 complex and the interaction with γ-TuSC. The requirement for the last 33 aa of Stu2 (*Figure 2a*) and for aa 116 to 230 of Spc72 (*Figure 1a*) for the Spc72–Stu2 interaction was confirmed in pulldown experiments (*Figure 2—figure supplement 1a and b*) (*Usui et al., 2003*). Quantitative MST measurements determined the $K_d$ of the Spc72N and Stu2 interaction as 115 nM (*Figure 2b*). This high-affinity binding was dependent on the Stu2-binding site of Spc72 and the C-terminus of Stu2, as Spc72N<sup>ΔStu2</sup> showed 23-fold reduced binding to Stu2 (*Figure 2b*) and because a Stu2 construct that lacks the 33 C-terminal aa (Stu2<sup>ΔC</sup>) did not interact with Spc72N (*Figure 2b*). Surprisingly, Spc72N<sup>CM1</sup>also reduced the binding affinity to Stu2 in the quantitative MST assay (*Figure 2b*). Thus, mutations in the CM1 motif of Spc72 not only affect the interaction of Spc72 with γ-TuSC (*Figure 1d*) but also Spc72 binding to Stu2, indicating an unexpected interdependency between both binding sites in Spc72.

To study the Spc72–Stu2 interaction with γ-TuSC, we developed a scheme for the purification of recombinant Spc72N–Stu2 complexes (*Figure 2—figure supplement 1c and d*). γ-TuSC bound to the pre-assembled Spc72N–Stu2 complex with 22-fold higher affinity than to Spc72N alone (compare *Figure 1d* and *Figure 2c*). Next, we asked whether direct binding of Stu2 to the γ-TuSC could be the reason for the enhanced interaction. Indeed, MST measurements indicated high-affinity binding of Stu2 to the γ-TuSC with a $K_d$ of 63.4 nM (*Figure 2d*). To understand which domain of Stu2 was responsible for this interaction with γ-TuSC, we used pulldown assays with the recombinant TOG domains of Stu2 (Stu2<sup>TOG</sup>; described in [*Widlund et al., 2012*]) and the fragment spanning the MT binding region of Stu2 (Stu2<sup>MT</sup>; aa 451–684). As expected, α/β-tubulin bound efficiently to the Stu2<sup>TOG</sup> construct (*Ayaz et al., 2012*; *Widlund et al., 2012*) but such binding was not observed for the purified γ-TuSC (*Figure 2—figure supplement 1e*; Tub4-blot). By contrast, Stu2<sup>MT</sup> pulled down the γ-TuSC (*Figure 2—figure supplement 1f*; Spc97-blot) but failed to interact with α/β-tubulin. Thus, we were able to show that the Stu2 protein probably enhances the Spc72–γ-TuSC interaction by binding to both Spc72 and γ-TuSC.

## Stu2 enhances the γ-TuSC oligomerization propensity of Spc72

Spc110 oligomerizes γ-TuSC into MT nucleation-competent rings dependent on its CM1 motif (*Kollman et al., 2010*; *Lin et al., 2014*). A shift of the γ-TuSC monomer towards earlier elution fractions in size exclusion chromatography is an indication of this activity (*Lin et al., 2014*). We were interested in whether Spc72 with the shorter CM1 had a similar impact on γ-TuSC. In comparison to Spc110N, Spc72N had only weak γ-TuSC oligomerization activity (compare *Figure 2e* with *Figure 2—figure supplement 2a and b*). This weak activity of Spc72N was further decreased in the case of Spc72N<sup>CM1</sup> (compare *Figure 2e* with *Figure 2—figure supplement 2c*). In marked contrast, the complex of Spc72N and wildtype (WT) Stu2 (Stu2<sup>WT</sup>; complex: Spc72N–Stu2<sup>WT</sup>) shifted a larger portion of the γ-TuSC into the Dextran blue marked void volume fractions than did Spc72N alone (*Figure 2e*). A Stu2 mutant protein carrying a single mutation in each of the two TOG domains (Stu2<sup>TOGAA</sup>; R200A, R519A, see *Figure 3c* for illustration) (*Ayaz et al., 2014*; *Geyer et al., 2015*) in complex with Spc72N also shifted γ-TuSC to the void volume fractions similarly to Spc72N–Stu2<sup>WT</sup> (*Figure 2—figure supplement 2d*). Analysis of the γ-TuSC–Spc72N and γ-TuSC–Spc72N-Stu2<sup>WT</sup> complexes that eluted in the void volume of the gel filtration column by negative staining electron microscopy identified circular γ-TuSC assemblies that were denser and more complex when Stu2 was present (*Figure 2—figure supplement 2e*). Thus, the Spc72N–Stu2 complex induces γ-TuSC oligomerization into assemblies.

Next, we tested whether Stu2 alone can also induce oligomerization of the γ-TuSC. Incubation of purified Stu2<sup>WT</sup> with γ-TuSC for 40 min on ice led to the formation of sedimentable Stu2<sup>WT</sup>–γ-TuSC oligomers (*Figure 2—figure supplement 2f*). Interestingly, this property of Stu2 was dependent on

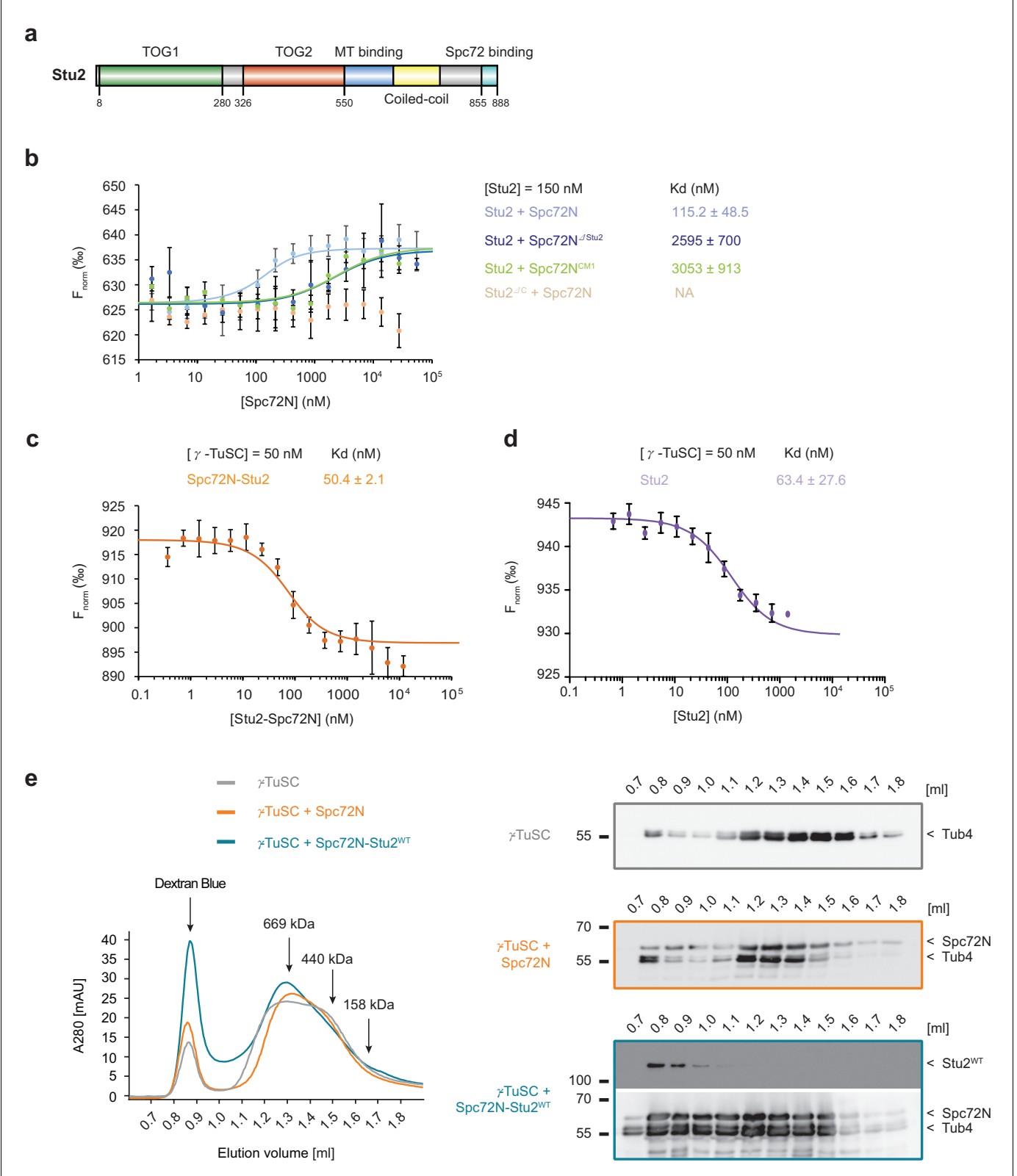

**Figure 2.** High-affinity binding of the Spc72N–Stu2 complex to γ-TuSC. (**a**) Scheme of the Stu2 substructure. Abbreviation: TOG, Tumor Overexpressed Gene. (**b**) MST measurements of Spc72N binding to Stu2. Mean ± SD, for Stu2 and Spc72$^{\Delta Stu2}$, n = 2; all others, n = 3; NA, not applicable. (**c**) High-affinity binding of Spc72N–Stu2 complexes to γ-TuSC as measured by MST. Mean ± SD, n = 3. (**d**) High-affinity binding of Stu2 to γ-TuSC as measured by MST. Mean ± SD, n = 3. Note, that the MST assay contained 0.05% Tween-20 to prevent precipitation of Stu2–γ-TuSC. (**e**) Spc72N–Stu2 induced

*Figure 2 continued on next page*

*Figure 2 continued*

oligomerization of γ-TuSC. The preassembled Spc72N–Stu2 complex shifted γ-TuSC to earlier elution fractions. The gel filtration profiles are shown relative to marker proteins (of 669, 440, 158 kDa). Dextran blue marks the void fraction of the chromatography column. Sample fractions were analysed by immunoblotting with the indicated antibodies (right). One representative example of three independent experiments is shown.

DOI: https://doi.org/10.7554/eLife.39932.003

The following figure supplements are available for figure 2:

**Figure supplement 1.** Spc72N but not Spc72N$^{\Delta Stu2}$ interacts with full-length Stu2.

DOI: https://doi.org/10.7554/eLife.39932.004

**Figure supplement 2.** γ-TuSC oligomerisation.

DOI: https://doi.org/10.7554/eLife.39932.005

its MT-binding region as a Stu2 mutant in which this domain was deleted (Stu2$^{\Delta MT}$; deletion of aa 550–658, see *Figure 3c* for illustration) did not have the same impact on γ-TuSC. This indicates that Stu2 can assemble γ-TuSC into larger assemblies dependent on its MT-binding site.

## Purified Stu2 in complex with Spc72 assembles MT asters

The data presented above indicate that Spc72–Stu2 forms a complex with γ-TuSC and oligomerizes γ-TuSC into circular assemblies. To understand the relative contribution of each of the components in MT formation, we analysed the MT nucleation activity of purified Stu2 and Spc72N and of the Spc72N–Stu2 complex with (*Figure 4*) and without (*Figure 3*) recombinant γ-TuSC in an in vitro MT nucleation assay. Fluorescently labelled α/β-tubulin was incubated with purified proteins either individually or in combination. The α/β-tubulin concentration of 12 µM was chosen because it enables the oligomerized Spc110N–γ-TuSC to nucleate microtubules (*Kollman et al., 2010*; *Lin et al., 2014*). A similar α/β-tubulin concentration has been used in MT nucleation assays with purified TPX2 and chTOG (*Roostalu et al., 2015*). Assembled MTs were spun onto cover slips and analysed by fluorescence microscopy.

Purified Spc72N did not have MT nucleation activity as it behaved similarly to the α/β-tubulin control (*Figure 3a*; see *Figure 4b* for quantification). The MT polymerase Stu2 occasionally nucleated MT asters (*Figure 3a and b*), but the overall MT nucleation activity of Stu2$^{WT}$ was low (*Figure 3—figure supplement 1b*). The Spc72N–Stu2$^{WT}$ complex promoted MT aster formation with greater frequency than did Stu2$^{WT}$ alone (*Figure 3a and b*), leading to an increase in MT nucleation activity (*Figure 4b*). The aster-forming activity was dependent on the preassembled Spc72N–Stu2$^{WT}$ complex as addition of the individual Spc72N and Stu2$^{WT}$ proteins to the reaction mixture did not promote aster formation beyond the activity of Stu2 alone (*Figure 3—figure supplement 1c*; Spc72N + Stu2$^{WT}$). Thus, the recombinant Stu2–Spc72$^{WT}$ complex predominately assembles MTs in the form of asters.

We next analysed the subdomains in Stu2 for their importance in MT nucleation. For this, mutant versions of Stu2 were constructed as outlined in *Figure 3c*. We confirmed that the recombinant Stu2$^{TOGAA}$ protein failed to bind α/β-tubulin in gel filtration and pulldown assays (*Figure 3—figure supplement 1d and e*) (*Ayaz et al., 2012*). Stu2$^{\Delta MT}$ and a Stu2 variant with a deletion of the dimerizing coiled-coil domain (Stu2$^{\Delta CC}$; deletion of aa 658–761, see *Figure 3c*) were constructed as reported previously (*Al-Bassam et al., 2006*; *Wang and Huffaker, 1997*). A yeast growth test indicated that all *STU2* sub-regions were important for viability. *stu2$^{\Delta MT}$* and *stu2$^{TOGAA}$* cells failed to grow at 23°C or 37°C. *stu2$^{\Delta CC}$* and *stu2$^{\Delta C}$* cells grew at 23°C but showed reduced growth at 37°C (*Figure 3—figure supplements 1f*, 5-FOA) and behaved in a dominant-negative manner at 37°C as indicated by growth retardation of cells carrying WT and mutant versions of *STU2* (*Figure 3—figure supplement 1f*; compare growth of cells at 23°C and 37°C on SC-Leu plates).

Next, Stu2 proteins were expressed and purified from insect cells either as a Stu2 version or in complex with Spc72N (*Figure 2—figure supplement 1c and d*). Stu2$^{TOGAA}$, Stu2$^{\Delta MT}$ and Stu2$^{\Delta CC}$ in combination with α/β-tubulin lacked the weak MT aster-forming ability of the Stu2$^{WT}$ (*Figure 3—figure supplement 1a and c*). In addition, Stu2$^{\Delta MT}$ and Stu2$^{\Delta CC}$ in complex with Spc72N did not assemble MT asters, as was the case for Spc72N–Stu2$^{WT}$ (*Figure 3b and d*). Interestingly, Spc72N–Stu2$^{TOGAA}$ assembled asters with the same frequency as Spc72N–Stu2$^{WT}$ (*Figure 3b and d*) that had a similar MT intensity (*Figure 3e and f*). However, the overall MT nucleation activity of Spc72N–Stu2$^{WT}$ was slightly higher than that of Spc72–Stu2$^{TOGAA}$ (*Figure 4b*). Thus, Stu2 in complex with

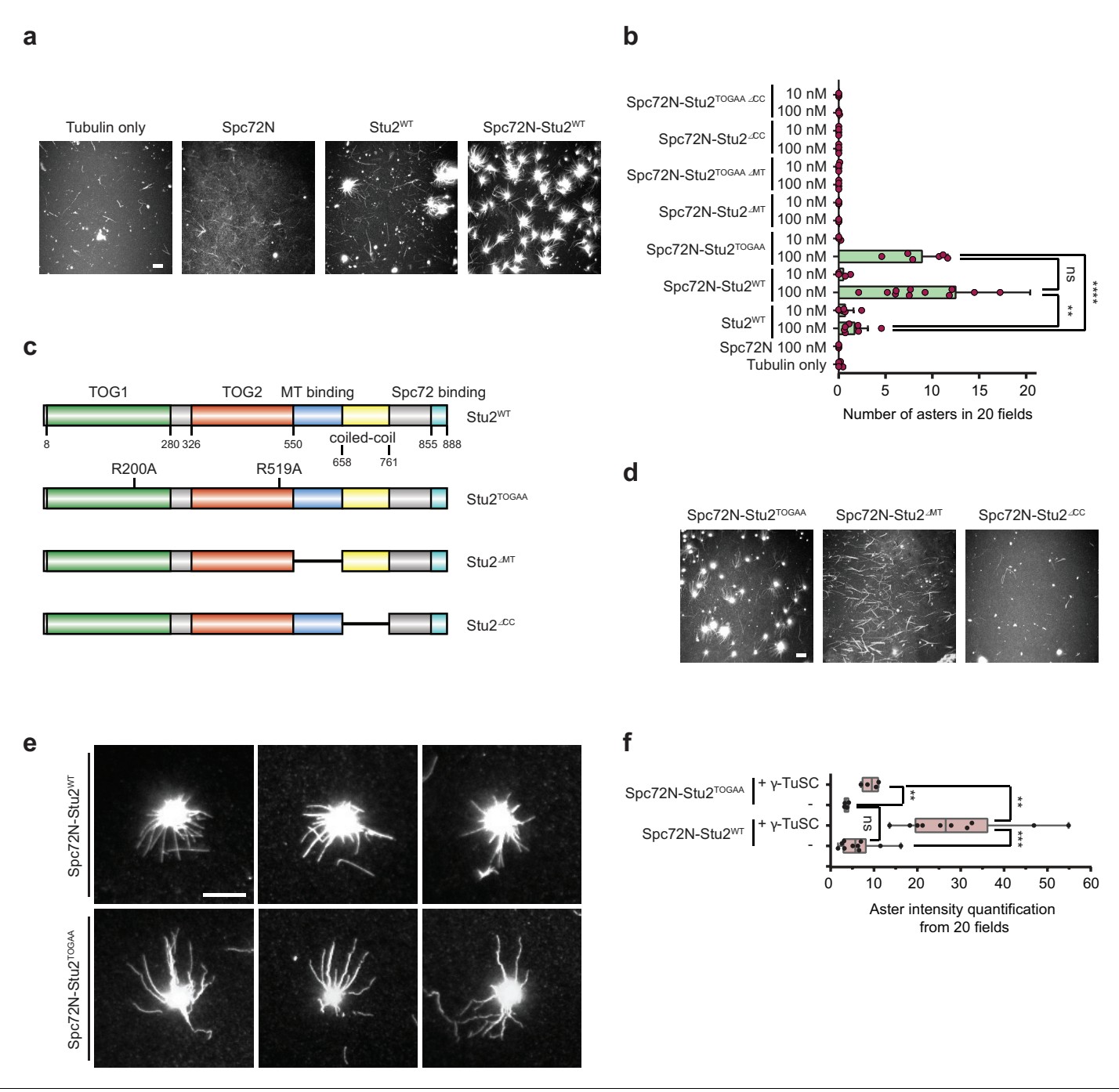

**Figure 3.** The purified Spc72N–Stu2 complex assembles MT asters. (**a**) In vitro MT assembly by purified Spc72N, Stu2 and the Spc72N–Stu2 complex. Representative images are maximum projections. Scale bar: 10 µm. (**b**) Quantification of aster numbers from (**a and d**) from 20 acquired regions per condition and a minimum of three independent experiments. Purple dots represent individual values. Values were corrected by the amount of nucleated MTs in the tubulin-only control. Mean ± SD; **p=0.0023, ****p<0.0001; ns, not significant. (**c**) Stu2-domain organization and mutant Stu2 proteins affecting the TOG1/2 domains (Stu2$^{TOGAA}$), the MT binding site (Stu2$^{\Delta MT}$), the coiled-coil region (Stu2$^{\Delta CC}$) and the C-terminus (Stu2$^{\Delta C}$). (**d**) In vitro MT assembly by purified Spc72N–Stu2$^{TOGAA}$, Spc72N–Stu2$^{\Delta MT}$ and Spc72N–Stu2$^{\Delta CC}$ complexes. Representative images are maximum projections. Scale bar: 10 µm. (**e**) Enlargement of representative MT asters from (**a and d**). Scale bar: 10 µm. (**f**) Aster density of the indicated complexes was analysed by quantifying the aster intensity from 20 fields of view.

DOI: https://doi.org/10.7554/eLife.39932.006

The following figure supplement is available for figure 3:

**Figure supplement 1.** MT aster-forming activity of Stu2 mutant proteins.

*Figure 3 continued on next page*

*Figure 3 continued*

DOI: https://doi.org/10.7554/eLife.39932.007

Spc72 assembles MT asters dependent on its MT-binding site and the dimerizing coiled-coil region. In this assay, mutations in the TOG domains of Stu2 did not affect the density of the MTs in the asters but the overall MT nucleation activity was slightly reduced.

As next a step, we measured the impact of γ-TuSC combined with Spc72N, Stu2 and Spc72N–Stu2 for MT nucleation. Incubation of α/β-tubulin with γ-TuSC led to the formation of single MTs. However, the nucleation activity was low (**Figure 4a and b**). The addition of Spc72N to γ-TuSC produced a similarly low MT nucleation activity (**Figure 4a and b**), probably reflecting the poor oligomerization of γ-TuSC by Spc72N (**Figure 2e**). Interestingly, γ-TuSC enhanced the MT nucleation and aster-forming activities of Stu2–Spc72N by factors of 3.1 and 1.5, respectively (**Figure 4a,b and c**). For this enhancement, Stu2 had to be in complex with Spc72N as the MT nucleation and aster-formation activity of Stu2 alone was not changed upon the addition of γ-TuSC (**Figure 3—figure supplement 1a** to c). Thus, γ-TuSC increases the ability of the Spc72–Stu2 complex to nucleate MTs and to form asters.

Subsequently, we tested the domains of Stu2 that were needed for MT nucleation together with γ-TuSC. Aster formation by Spc72N–Stu2–γ-TuSC complex was dependent on the dimerizing coiled-coil region and the MT-binding domain of Stu2 (**Figure 4a and b**). The two TOG domains of Stu2 did not influence the number of MT asters formed by Spc72N–Stu2–γ-TuSC (**Figure 4a and c**). However, MT asters, which assembled in the presence of Spc72N–Stu2–γ-TuSC, were denser than those formed by Spc72N–Stu2$^{TOGAA}$-γ-TuSC (**Figure 3f** and **Figure 4d**). This reduction in aster density was reflected in a lower overall MT nucleation activity of Spc72N–Stu2$^{TOGAA}$-γ-TuSC compared to that of Spc72N–Stu2$^{WT}$-γ-TuSC (**Figure 4b**). As Spc72N–Stu2$^{TOGAA}$ efficiently assembled γ-TuSC oligomers (**Figure 2—figure supplement 2d**), it is likely that the reduction of the MT density in the Spc72N–Stu2$^{TOGAA}$-γ-TuSC asters is caused by a decrease in the MT polymerization activity of Stu2$^{TOGAA}$.

This analysis indicates that Spc72N–Stu2 efficiently assembles MTs in cooperation with γ-TuSC. The TOG domains, the MT-binding site and the dimerizing coiled-coil region of Stu2 are required for the full MT nucleation activity of the Spc72N–Stu2–γ-TuSC complex.

## Spc72 and Kar9 target Stu2 to the cMT minus end

We next used a series of in vivo experiments to test the role of Stu2 in maintaining γ-TuSC at the minus end of detached cMTs (**Figure 5** and **Figure 6**) and the function of Stu2 and Tub4 in cMT nucleation (**Figure 7**). Because the nuclear export of Stu2 is cell-cycle regulated (**van der Vaart et al., 2017**), we performed these experiments in α-factor-arrested G$_1$ cells when Stu2 is mainly cytoplasmic to exclude cell-cycle effects.

In order to obtain detached cMTs with a bound MT nucleation site at the minus end, we treated *kar1Δ15* cells with α-factor. In the presence of α-factor, it is the Kar1–Spc72 interaction that anchors cMTs to the bridge structure of the SPB (**Pereira et al., 1999**). The mutated Kar1Δ15 protein lacks the region that mediates Spc72 binding. Therefore, in around 40% of *kar1Δ15* cells, cMTs detach from the SPB into the cytoplasm in response to α-factor treatment (**Figure 5a**). Previously, we have shown by electron tomography that detached cMTs are single MTs with a Spc72–γ-TuSC cap at their minus end (**Erlemann et al., 2012**). We noticed that in around 70% of cells, the minus end of detached cMTs (as marked by Spc72 or Tub4) localized within the schmoo tip region (2 µm radius). This orientation is probably a reflection of the association of the minus end motor Kar3 with the schmoo tip (**Maddox et al., 2003**), pulling the detached cMT minus ends into the schmoo.

We first used this system of detached cMTs to study the localisation of Stu2 at the cMT minus end. In 85% of *SPC72 kar1Δ15* cells with detached cMTs, Stu2–yeGFP was enriched together with Spc72–mCherry at the minus end of cMTs (**Figure 5b**, top). About 15% of detached cMTs did not carry a Stu2 peak signal that overlapped with Spc72 (**Figure 5b**, bottom). Stu2–yeGFP weakly decorated the MT wall of both detached cMT types (**Figure 5b**). Surprisingly, Stu2–yeGFP also associated with the minus end marker Tub4–mCherry in the majority (61%) of *spc72$^{ΔStu2}$ kar1Δ15* cells with detached cMTs (**Figure 5c**, top), whereas 39% of detached cMTs did not show Stu2 enrichment with

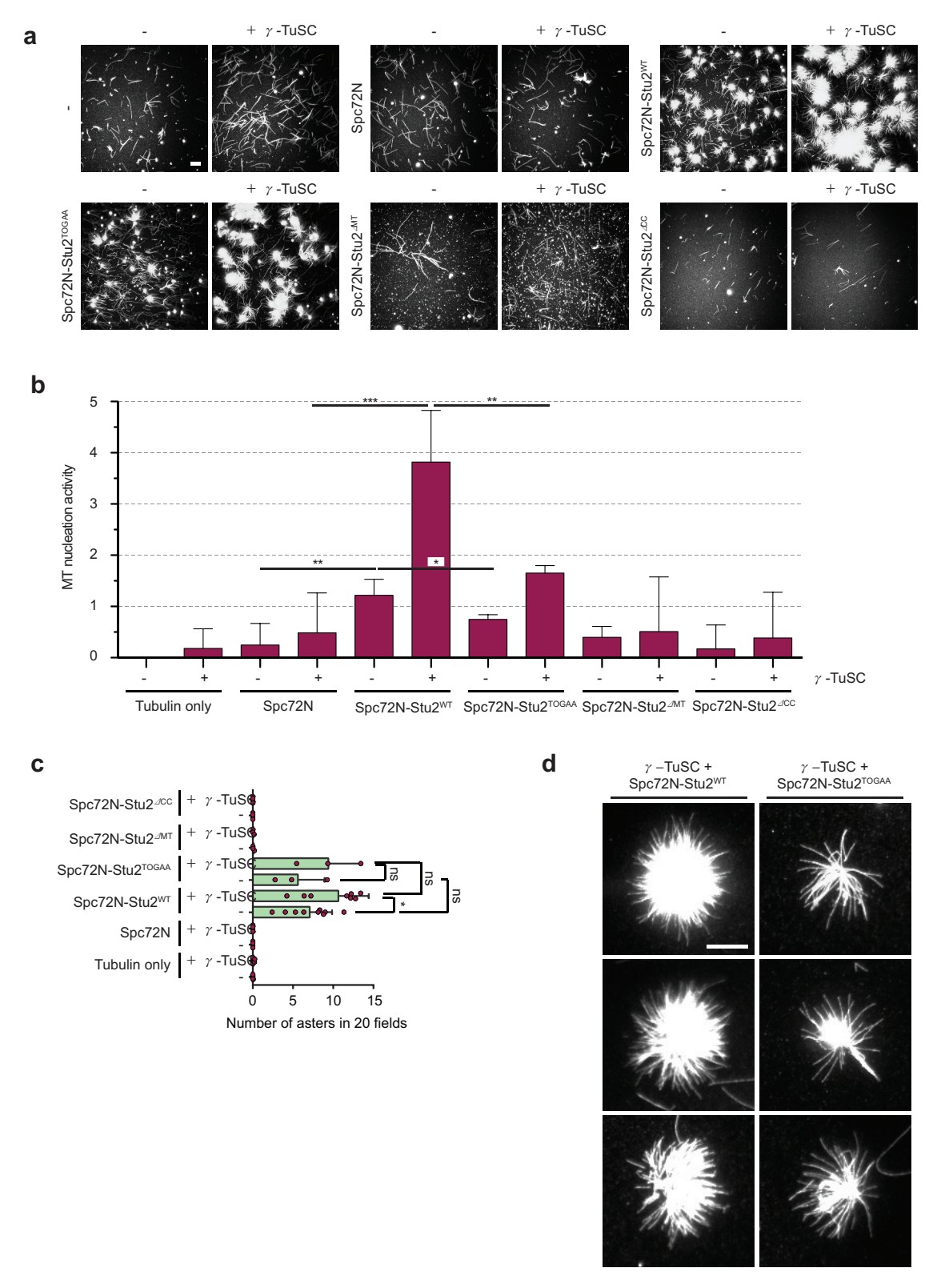

**Figure 4.** The γ-TuSC together with Spc72N–Stu2 increases aster intensity and number. (a) In vitro MT assembly reactions as in *Figure 3* with the indicated purified proteins. Representative images are maximum projections. Scale bar: 10 μm. (b) Quantification of MT nucleation activity illustrated in (a) as described in the 'Materials and methods'. The tubulin-only background was subtracted from all samples. Three independent experiments; mean ± SD; *p≤0.03, **p=0.01, ****p<0.001. (c) Quantification of the aster number of samples illustrated in (a) from 20 acquired regions per condition

*Figure 4 continued on next page*

Figure 4 continued
in each experiment and a minimum of three experiments. Values were corrected by the number of nucleated MTs in the tubulin-only control.
Mean ± SD, purple dots represent the values of individual experiments; *p≤0.0367; ns, not significant. (d) Enlargement of representative MT asters from the samples illustrated in (a). Scale bar: 10 µm.
DOI: https://doi.org/10.7554/eLife.39932.008

Tub4 (*Figure 5c*, bottom). Spc72$^{\Delta Stu2}$ still localized to the end of detached cMT (*Figure 6—figure supplement 1a*). The Spc72–yeGFP and Stu2–yeGFP signals at the minus end of detached cMTs were quantified in order to gain a better understanding of their recruitment to this site. The normalized signal intensity at the cMT minus end was 0.19 for Spc72–yeGFP and 0.95 for Stu2–yeGFP in WT *kar1Δ15* cells (*Figure 5d and e*). In *spc72$^{\Delta Stu2}$ kar1Δ15* cells, the normalized intensity was 0.17 for Spc72$^{\Delta Stu2}$-yeGFP and 1.25 for Stu2–yeGFP (*Figure 5d and e*). Together, this evidence indicates a five-fold excess of Stu2 over Spc72 at cMT nucleation sites, as well as Spc72-independent recruitment of Stu2 to the cMT minus end.

The existence of another Stu2 interaction partner at the cMT minus end was further supported by fluorescence recovery after photobleaching (FRAP) experiments. Spc72–yeGFP at detached cMTs showed no recovery after bleaching. By contrast, 58% of Stu2–yeGFP recovered with a half time of 3.3 s, indicating a dynamic and stable Stu2 pool (*Figure 5f*). In *spc72$^{\Delta Stu2}$ kar1Δ15* cells, Stu2–yeGFP at cMTs ends almost completely recovered with a half time of 5.6 s. The complete recovery in the latter experiment also ruled out the possibility of the limitation of freely available Stu2–yeGFP, as the analyzed *spc72$^{\Delta Stu2}$ kar1Δ15* cells showed higher Stu2–yeGFP signal intensity at the cMT minus end before bleaching in comparison to *SPC72 kar1Δ15* (*Figure 5f*). In conclusion, a stable fraction of Stu2 binds to the minus end of cMTs via Spc72 and a dynamic fraction binds via an additional factor.

## Stu2 stabilizes γ-tubulin at cMT ends

We wanted to identify the second recruitment factor for Stu2 at the minus end of cMTs. A possible candidate was the MT-associated protein Kar9, which localises to the SPB in a process that is dependent on the C-terminus of γ-tubulin Tub4 (*Cuschieri et al., 2006*). In addition, Kar9 is a known interaction partner of Stu2 (*Miller et al., 2000*; *Moore and Miller, 2007*). Consistent with these interactions, Kar9 co-localized dot-like with Spc72–mCherry at 81% of detached cMTs (*Figure 6a* and *Figure 6—figure supplement 1b*). Quantification of the Kar9–yeGFP single intensity at the minus end of detached cMTs indicated a high relative abundance of 1.79 in *SPC72 kar1Δ15* cells (normalized to Cse4–yeGFP) that was even further increased to 3.27 in *spc72$^{\Delta Stu2}$ kar1Δ15* cells (*Figure 6—figure supplement 1c*). Thus, the lack of the Stu2-binding site in Spc72 increases Kar9 recruitment to the cMT minus end. Considering Stu2 binding to Kar9 (*Miller et al., 2000*; *Moore and Miller, 2007*), this result provides an explanation for the enhanced binding of Stu2 to the cMT minus end in *spc72$^{\Delta Stu2}$* cells (*Figure 5e*).

To show that a fraction of Stu2 is indeed recruited to the cMT minus end via Kar9, we constructed endogenous *KAR9* gene fusions with the auxin-induced degron (AID)-tag (*Nishimura et al., 2009*). Addition of the auxin homologue IAA to *KAR9–AID kar1Δ15* cells promoted the degradation of Kar9–AID (*Figure 6—figure supplement 1d*) and decreased Stu2–yeGFP intensity at cMT minus ends from 0.95 to 0.44 (*Figure 6b*). By contrast, Spc72 was not affected by the depletion of Kar9 (*Figure 6c*). These data support the notion that Kar9 is the second Stu2-binding factor at the cMT minus end. Therefore, the depletion of Kar9 in *spc72$^{\Delta Stu2}$* cells should further decrease Stu2–yeGFP intensity at the minus end of detached cMTs. However, as outlined below, these measurements were complicated by the instability of cMTs in *spc72$^{\Delta Stu2}$ KAR9–AID kar1Δ15* cells and by the observation that of the surviving detached cMTs only half carried Tub4 (*Figure 6f and g*). The normalized Stu2–yeGFP signal intensity at these rare Tub4-positive cMT minus ends was measured with ~0.82, which is clearly higher than the signal in *KAR9–AID* cells (0.44; compare *Figure 5e* with *Figure 6b*). This suggests that Kar9–AID was not fully degraded in these *spc72$^{\Delta Stu2}$ KAR9–AID* cells with detached cMTs. Indeed, variations between yeast cells in the efficiency of the auxin-induced degradation of proteins have been reported (*Papagiannakis et al., 2017*). This residual amount of Kar9–AID in the small number of *spc72$^{\Delta Stu2}$ KAR9–AID* cells with cMTs was, however, below the detection

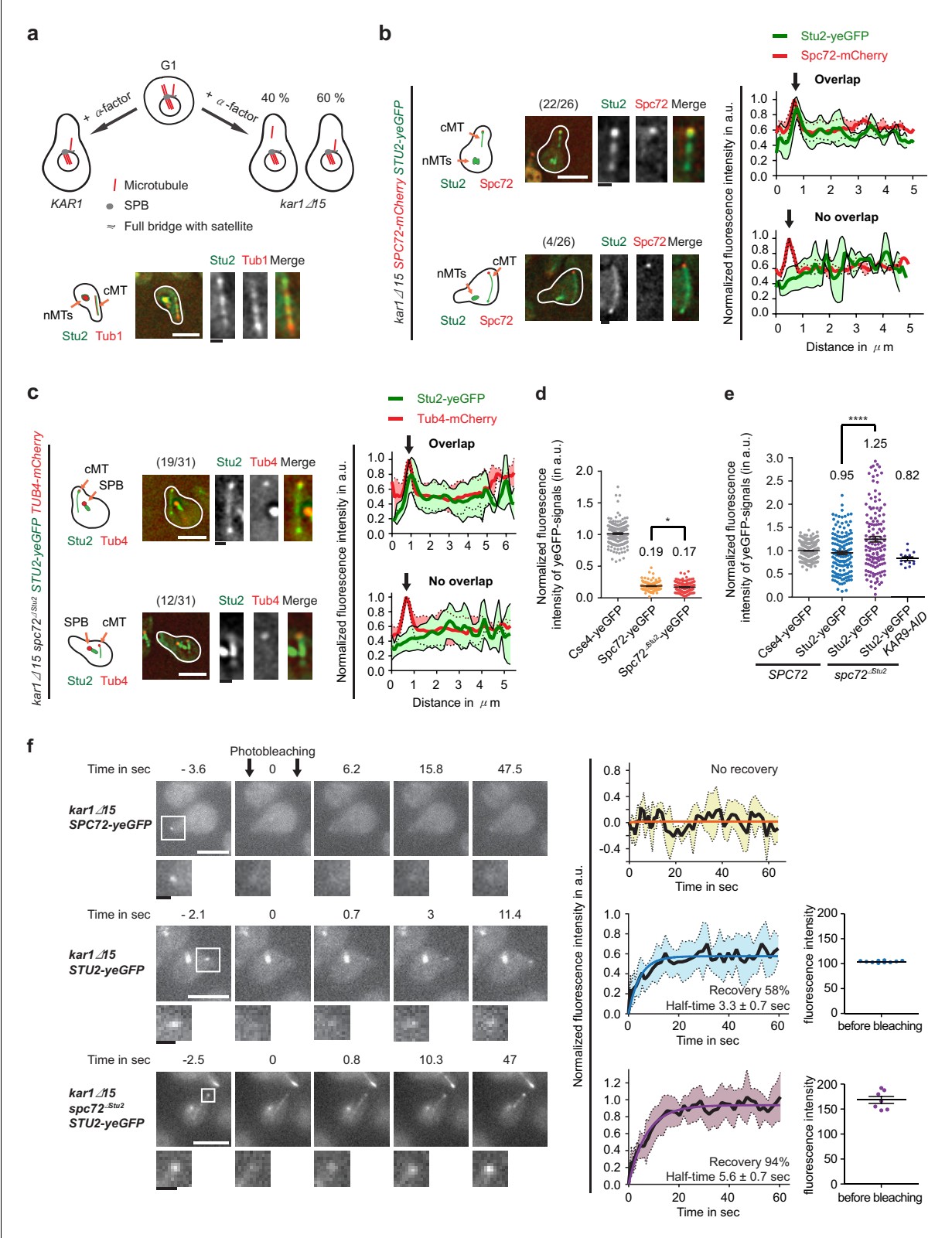

**Figure 5.** Stu2 associates to the minus end of detached cMTs via Spc72 and an additional factor. (a) Model for the generation of detached cMTs. G1 cells organize cMTs from the SPB outer plaque and the bridge structure. Upon addition of α-factor, the bridge organzies cMTs by binding of γ-TuSC–Spc72 to the bridge component Kar1. In about 40% of *kar1Δ15* cells, this interaction is lost and cMTs are released into the cytoplasm with γ-TuSC and Spc72 at the MT minus end. A single stack of an exemplary cell is shown at the bottom. The cartoon on the left illustrates the position of the detached

*Figure 5 continued on next page*

*Figure 5 continued*

cM in the cell on the right. White scale bar: 5 µm; black scale bar: 1 µm. (b–f) Experiments were performed with *kar1Δ15* cells treated with α-factor and analysis was performed on single, detached cMTs. (b) Stu2–yeGFP associates with the minus end of detached cMTs. Numbers in brackets represent cells with this phenotype/total cell number. Representative images are single z-planes. The graphs on the right show line scans of the Stu2 signal (green) and the Spc72 signal (red) along the cMTs. The line represents the mean and the surrounding area the SD of the measured signals. The arrow marks the cMT minus end. White scale bar: 5 µm, black scale bar: 1 µm. (c) As in (b): Stu2–yeGFP associates with the majority of the minus ends of detached cMTs in *spc72^{ΔStu2}* cells. (d) Normalized intensity of Spc72 and Spc72^{ΔStu2} at detached cMTs. The kinetochore protein Cse4 from late anaphase cells was used as a fluorescence intensity standard. Mean ± SEM, n ≥ 129 cells. *p=0.0319. (e) Normalized intensity of Stu2 at detached cMT minus ends in *SPC72*, *spc72^{ΔStu2}* and *spc72^{ΔStu2} KAR9-AID* (1 hr IAA) cells. Cse4 was used as a fluorescence standard. Mean ± SEM, n ≥ 134 cells, ****p<0.0001. (f) Analysis of the dynamic behaviour of Spc72–yeGFP and Stu2–yeGFP at detached cMT minus ends by fluorescence recovery after photobleaching (FRAP). Representative images are from single z-planes with enlargement of the bleached signal shown underneath. Corresponding graphs on the side represent the normalized amount of signal recovery and the fluorescence intensity of the Stu2–yeGFP signal at the beginning of the experiment before bleaching. N(Spc72) = 6 cells; n(Stu2) = 10 cells each. Size bars are as in (b).
DOI: https://doi.org/10.7554/eLife.39932.009

limit of the immunoblot in *Figure 6—figure supplement 1d*. Taken together, this evidence suggests that Kar9 and Spc72 recruit Stu2 to the minus end of cMTs.

Subsequently, we aimed to analyse the function of Stu2 at the minus end of detached cMTs. We used *KAR9–AID* in combination with the *spc72^{ΔStu2}* mutation in order to reduce Stu2 specifically at the minus end of cMTs. *kar1Δ15* cells were incubated with α-factor for 60 min simultaneously with the addition of IAA or the solvent control. *spc72^{ΔStu2} KAR9–AID kar1Δ15* cells that were exposed to IAA showed a reduction in cMT numbers (33%; attached and detached) in comparison to the *SPC72 KAR9–AID kar1Δ15* cells (*Figure 6d*), indicating a function for Stu2 in the nucleation or stability of cMTs.

The C-terminal 33 aa of Stu2 mediate binding with Spc72 and Kar9 (*Moore and Miller, 2007*; *Usui et al., 2003*). Therefore, we would expect that *stu2^{ΔC} kar1Δ15* cells would show a reduction in cMT number similar to that shown by *spc72^{ΔStu2} KAR9–AID kar1Δ15* cells that have been exposed to IAA. This was indeed the case (*Figure 6—figure supplement 1e*). Using immunoblotting, we confirmed that Stu2^{ΔC} was expressed similarly to WT Stu2 (*Figure 6—figure supplement 1f*). Together, these experiments led us to conclude that the impairment of both Stu2-binding sites at the cMT minus end affects cMT stability or nucleation.

To further clarify the role of Stu2 in cMT function, we performed a time course experiment with *spc72^{ΔStu2} KAR9 kar1Δ15* cells and *spc72^{ΔStu2} KAR9–AID kar1Δ15* cells (*Figure 6e*). In this experiment, Kar9 was depleted after detachment of cMTs from the SPB, meaning that we analysed the function of Stu2 on the preassembled γ-TuSC oligomere. We observed that the number of detached cMTs in *spc72^{ΔStu2} KAR9–AID kar1Δ15* cells gradually decreased from 50% (t = 0) to 25% within 60 min after the addition of IAA (*Figure 6f*). This decrease coincided with a general decline in cMT-number in *spc72^{ΔStu2} KAR9–AID kar1Δ15* cells in the presence of IAA from 83% to 27%. Thus, Stu2 at the minus end of detached cMTs is probably needed for their stability.

One possible interpretation of the outcome of the experiment in *Figure 6f* is the detachment of the γ-TuSC cap, followed by MT depolymerisation when Stu2 is removed from the cMT minus end. Indeed, IAA addition to *spc72^{ΔStu2} KAR9–AID kar1Δ15* cells reduced not only the number of cMTs but also the percentage of detached cMTs with a Tub4 signal from close to 100% to 54% (*Figure 6g*). This analysis suggests that Stu2 removal from the cMT minus end triggers the disassembly of γ-TuSC cap followed by cMT depolymerisation. It is in agreement with the in vitro data illustrated in *Figure 2*, indicating a stabilizing role for Stu2 in the Spc72–γ-TuSC interaction.

## MT nucleation with reduced γ-tubulin Tub4

*Figure 6* has shown a function of Stu2 in maintaining the γ-TuSC oligomer at the minus end of assembled cMTs. To judge the impact of Tub4 and Stu2 on MT nucleation, we developed an in vivo cMT nucleation assay using *TUB4–AID* or *STU2–AID* degron cells with the SPB marker *SPC42–mCherry* and the MT marker *GFP–TUB1*.

We first analysed the role of Tub4 in cMT nucleation. The addition of IAA to *TUB4–AID* cells induced the degradation of Tub4–AID below the critical threshold that was needed for viability (*Figure 7—figure supplement 1a*). However, we noticed that ~10% of Tub4–AID resisted degradation

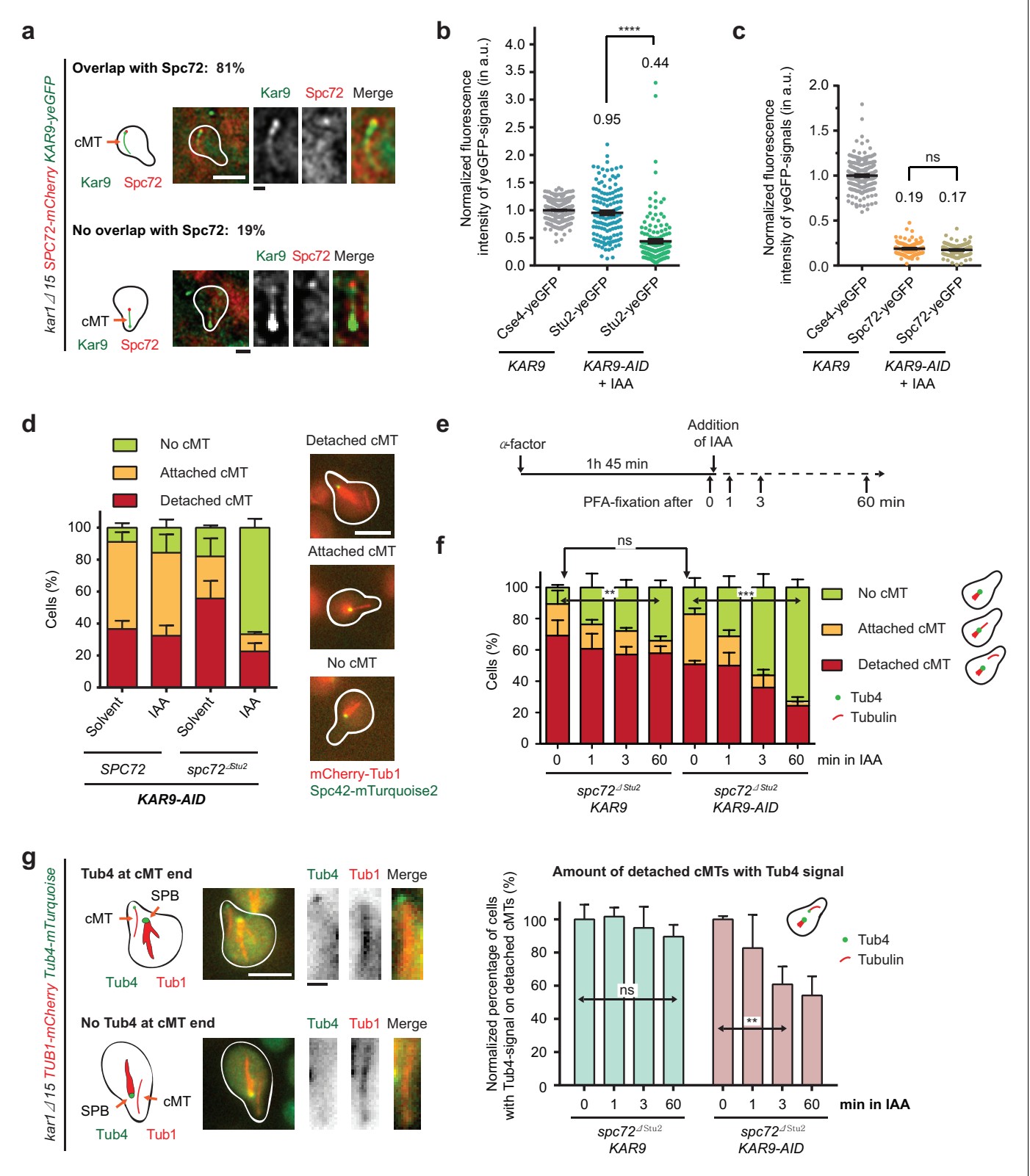

**Figure 6.** Kar9-dependent recruitment of Stu2 to the cMT minus end. (**a**) Kar9–yeGFP localization on detached cMTs. Spc72–mCherry marks the cMT minus end. Representative images are from single z-planes. N = 27 cells; white scale bar — 5 μm; black scale bar — 1 μm. (**b**) Normalized intensity of Stu2–yeGFP at detached cMT minus ends in WT cells or after auxin-induced depletion of Kar9 for 1 hr. The cMT minus end was marked by Tub4–mCherry. Cse4 was used as a fluorescence intensity standard. Note, the Stu2–yeGFP value is the same as in *Figure 5e*. Strains were analysed

*Figure 6 continued on next page*

*Figure 6 continued*

simultaneously. Mean ± SEM, n ≥ 140 cells for each condition. ****p<0.0001. (c) Normalized intensity of Spc72–yeGFP at the cMT minus end of mCherry–Tub1 labelled detached cMTs. Measurements were done in *KAR9 kar1Δ15* cells and *KAR9-AID kar1Δ15* cells with auxin-induced depletion of Kar9 for 1 hr. The kinetochore signal of *CSE4–yeGFP* cells was used as a fluorescence intensity standard. Note, the Spc72–yeGFP value is the same as that in *Figure 5d*. Strains were analysed simultaneously. Mean ± SEM, n ≥ 104 cells; ns, not significant. (d) Indole-3-acetic acid (IAA) or the solvent ethyl alcohol (EtOH) were added for 1 hr to α-factor-arrested *SPC72* and *spc72^ΔStu2* cells containing *KAR9-3HA-AID*. Mean ±SD, n ≥ 153 cells in three independent experiments. Scale bar: 5 μm. (e) Scheme of the experiment shown in (f). The solvent is EtOH. (f) Number of cMTs in *spc72^ΔStu2^ KAR9-3HA kar1Δ15* or *spc72^ΔStu2^ KAR9-3HA-AID kar1Δ15* cells over time. Mean ± SD, n ≥ 50 cells in three independent experiments. **p=0.001, ***p≤0.0006; ns, not significant. (g) The combined impairment of Kar9 and the Stu2 binding site in *spc72^ΔStu2^* triggers the detachment of Tub4–mTurquoise2 from the minus end of detached cMTs. Conditions as in (e). Cells were fixed 0, 1, 3 and 60 min after IAA addition. The scale was normalized to the time point 0 min (=100%). Mean ± SD, n ≥ 50 cells; three independent experiments, **p=0.0033; ns, not significant. On the right: example cells with mCherry–Tub1 and with or without Tub4–mTurquoise2 signal are shown together with schematic drawings of the detached cMT. White scale bar — 5 μm; black scale bar — 1 μm.

DOI: https://doi.org/10.7554/eLife.39932.010

The following figure supplement is available for figure 6:

**Figure supplement 1.** Role of Stu2 at the cMT minus end.

DOI: https://doi.org/10.7554/eLife.39932.011

(*Figure 7—figure supplement 1b*). Next, *TUB4* and *TUB4–AID* cells were arrested with α-factor in G₁ followed by the addition of nocodazole to depolymerize MTs and of IAA to induce degradation of Tub4–AID. Finally, nocodazole was washed out to promote MT re-nucleation (*Figure 7a*). In all strain backgrounds, close to 100% of the cells invariably lacked MTs upon nocodazole treatment (*Adams and Pringle, 1984*; *Gombos et al., 2013*). In response to nocodazole wash out, MTs re-polymerized in WT *TUB4* cells as analysed after 30 min (*Figure 7b*). Only the nuclear MTs over-lapped with the NLS–BFP (the nuclear localization signal fused to a blue fluorescence protein) stain-ing region, whereas cMTs did not (*Figure 7b*), so we were able to distinguish the two MT-types clearly. About 97% of WT cells carried cMTs after 30 min of nocodazole wash out (*Figure 7b*). Most (86%) of these cMTs were associated with the Spc42–mCherry-marked SPB. Surprisingly, IAA-induced depletion of Tub4 only mildly affected cMT nucleation efficiency as 90% of the cells carried cMTs after 30 min (*Figure 7b*). Strikingly, about 45% of these cMTs were not associated with the SPB but resided in the cytoplasm away from the NLS–BFP-marked nucleus. Analysis of the localiza-tion of the γ-TuSC receptor Spc72–mCherry, and of the γ-TuSC complex components Spc97–mCherry and Spc98–mCherry in *TUB4–AID* cells after nocodazole wash out, indicated that the free cMTs did not contain Spc72, Spc97 or Spc98 (*Figure 7c*, left). In a control experiment, we estab-lished that these proteins were detected at the minus end of detached cMTs of *kar1Δ15* cells (*Figure 7c*, right) (*Erlemann et al., 2012*). Thus, our detection system was sensitive enough to visual-ize Spc72, Spc97 and Spc98 at the minus end of single cMTs. These data suggest γ-TuSC-indepen-dent cMT nucleation in *TUB4–AID* cells.

The nuclear MTs at the SPB of *TUB4–AID* cells were, however, associated with Spc97 and Spc98 (*Figure 7c*). The residual Tub4 in these cells (*Figure 7—figure supplement 1b*) was probably suffi-cient to assemble γ-TuSC, which then became concentrated in the nucleus by the nuclear localization signal in Spc98 (*Pereira et al., 1998*).

It was important to exclude the possibility that detached cMTs of *TUB4–AID* cells nucleate first at the SPB and then detach from this localization. For this, we performed live-cell image analysis of cells after nocodazole wash out. In Tub4-depleted *TUB4–AID SPC42–mCherry GFP–TUB1* cells, cMTs were assembled spontaneously away from the SPB (*Videos 1* and *2*). By contrast, in *TUB4* WT cells, cMTs formed at the Spc42–mCherry marked SPB (*Video 3*). This analysis eliminates the possibility of cMT formation at the SPB in *TUB4–AID* cells followed by their detachment.

To judge the efficiency of cMT assembly in Tub4-depleted cells, we compared the kinetics of cMT formation in WT *TUB4* and *TUB4–AID* cells using the live-cell imaging data. Upon nocodazole wash out, *TUB4–AID* cells assembled cMTs with kinetics and efficiency that were similar to those observed in WT cells, with the difference that the majority of cMTs in *TUB4–AID* cells were not associated with the SPB (*Figure 7d*; note that the free cMTs of *TUB4–AID* cells were more frequent in the MatTek dish live-cell imaging experiment than in the flask experiment in *Figure 7b*). Taken together, these

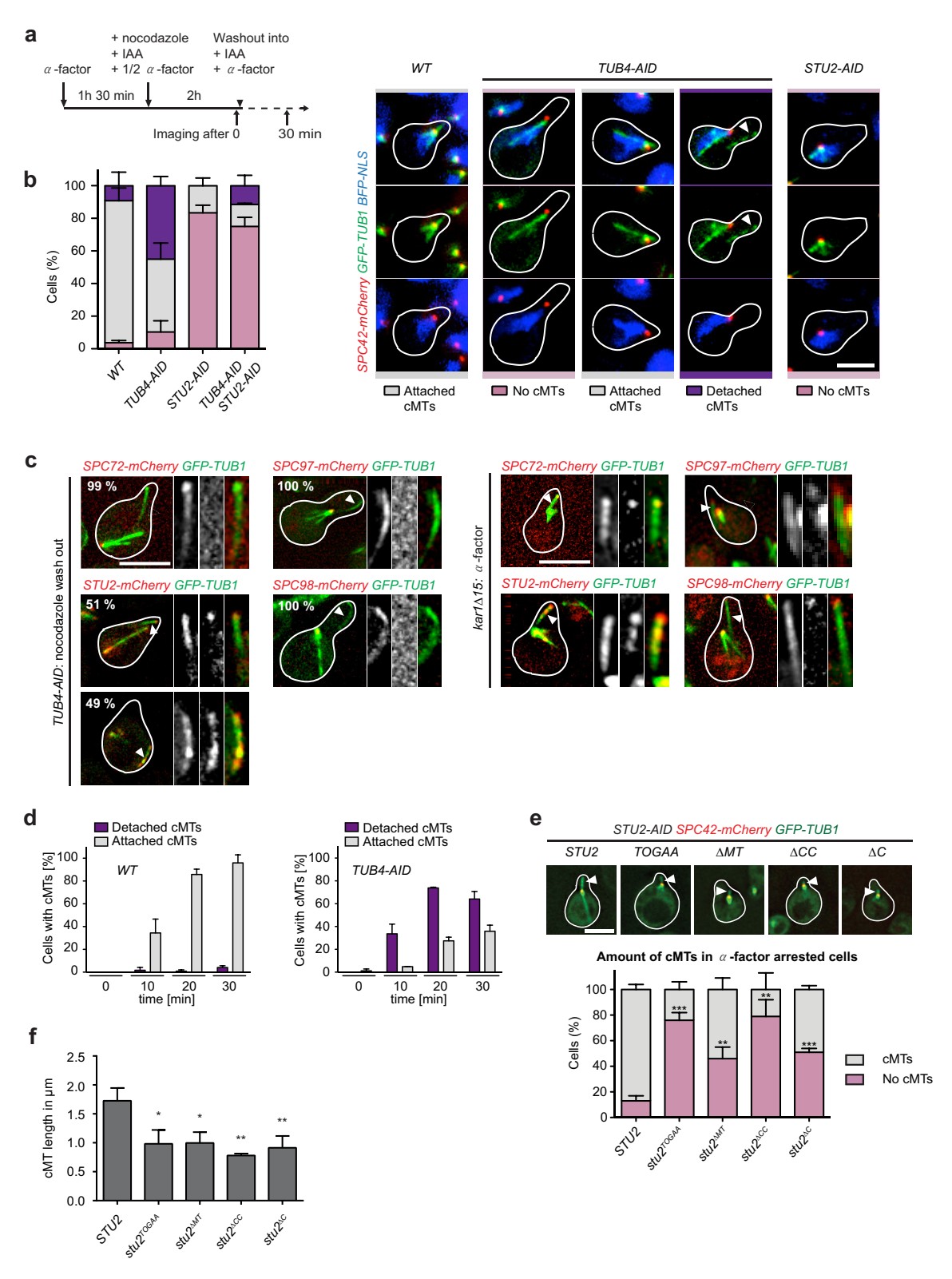

**Figure 7.** Tub4 limits cMT nucleation at the SPB while Stu2 is required for MT assembly in vivo. (**a**) Scheme of experiments in (**b and c**). (**b**) cMT assembly after nocodazole wash out as outlined in (**a**). Shown are representative cells with *yeGFP–TUB1 SPC42–mCherry NLS–BFP* that were used for the quantification summarized in the barchart on the left. Mean ± SD, n ≥ 30 cells per strain, per experiment. The experiment was repeated three times. The arrowheads in the *TUB4–AID* cells point towards cMTs in the cytoplasm. Scale bar: 10 μm. (**c**) *TUB4–AID yeGFP–TUB1* cells with the

*Figure 7 continued on next page*

Figure 7 continued

indicated mCherry markers were cultured as outlined in (**a**). The images show representative cells with cMTs that were not attached to the SPB (marked by arrowheads). The numbers indicate the percentage of cells demonstrating the illustrated phenotype (100 cells were analysed per condition). As control for the Spc72–, Stu2–, Spc97– and Spc98–mCherry signals on single detached cMTs, we incubated *kar1Δ15 GFP–TUB1* cells with α-factor for 60 min. Detached cMTs were analysed (arrowheads). The experiment was repeated twice with identical outcomes. (**d**) Kinetics of cMT assembly. *TUB4* WT and *TUB4–AID* cells were treated with α-factor and nocodazole as outlined in (**a**). cMTs were analysed 0, 10, 20 and 30 min after nocodazole washout by live cell analysis (see Videos 1–3). Under this experimental condition, *TUB4–AID* cells nucleated more detached cMTs compared to those shown in (**b**). Mean ± SD, n ≥ 30 cells, two independent experiments. (**e**) *STU2–AID yeGFP–TUB1 SPC42–mCherry* cells with the indicated integrated *STU2* constructs. The experiment was performed as outlined in (**a**). Three independent experiments were performed; n > 30 cells per experiment. **p<0.0039; ***p<0.0002. Example cells with long (*STU2* WT) or short cMTs (*STU2* mutants) are shown. Pictures are from single z-stacks with an arrowhead pointing towards the cMT. Scale bar: 5 µm. (**f**) Length of cMTs of cells from (**e**). Three experiments were performed; n > 30 cells per experiment.* p<0.0166; **p<0.0095.

DOI: https://doi.org/10.7554/eLife.39932.012

The following figure supplement is available for figure 7:

**Figure supplement 1.** The TOG domains of Stu2 are important for the nucleation of cMTs.
DOI: https://doi.org/10.7554/eLife.39932.013

experiments indicate that cells can efficiently nucleate cMTs without the involvement of the receptor Spc72 and the γ-TuSC. However, Tub4 activity is important to direct cMT nucleation to the SPB.

## Stu2 is essential for cMT nucleation

The in vitro analysis (*Figures 1–3*) has shown the ability of Stu2 to nucleate MTs even without γ-TuSC. Thus, Stu2 could be the nucleating factor that promotes cMT formation in cells with strongly reduced Tub4 levels. To assess this possibility, we asked whether the cMTs in Tub4-depleted cells require Stu2 for formation using the assay described in *Figure 7a* and whether Stu2 associates with detached cMTs in *TUB4–AID* cells. As hypothesized, cells with double depletion of Tub4 and Stu2 (*TUB4–AID STU2–AID* cells) failed to assemble cMTs (*Figure 7b* and *Figure 7—figure supplement 1b*). In addition, Stu2 localization analysis indicated that 51% of the free cMTs in *TUB4–AID* cells carried a strong enrichment of Stu2–mCherry at one end (*Figure 7c*, left). Frequently, this was the end in the schmoo tip region. The other detached cMTs (49%) were decorated with Stu2–mCherry. This evidence raises the possibility of a function of Stu2 in the assembly of cMTs when Tub4 levels are low.

We then asked whether Stu2 is also important for cMT nucleation in cooperation with γ-TuSC. To test this notion, we analysed the impact of Stu2–AID depletion on the nucleation of cMTs in *TUB4* cells using the assay described in *Figure 7a*. IAA depleted the Stu2–AID protein below the critical threshold that is needed for viability (*Figure 7—figure supplement 1a and b*). Stu2 depletion dramatically impaired the formation of cMTs without affecting the assembly of nuclear MTs (*Figure 7b*). This phenotype was complemented by an integrated WT *STU2* construct (*Figure 7e*). Thus, Stu2 is important for cMT nucleation in vivo in the presence of γ-TuSC.

Finally, we tested the role of Stu2 domains (*Figure 2a*) on cMT assembly in vivo. STU2 WT and mutant versions were expressed in *STU2–AID* cells. Upon Stu2–AID depletion, *stu2^{TOGAA}* was expressed as WT *STU2* (*Figure 7—figure supplement 1c*). The expression of *stu2^{ΔMT}* and *stu2^{ΔC}* was, in fact, slightly higher than that of WT *STU2*. *stu2^{ΔCC}* showed lower expression levels compared to *STU2*. *STU2–AID STU2* cells nucleated SPB-associated cMTs after nocodazole wash out (*Figure 7e*) as efficiently as WT *STU2* cells (*Figure 7b*, WT). These cMTs were ~1.8 µm long and connected the SPB with

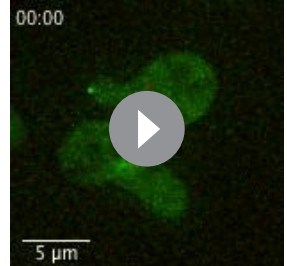

**Video 1.** *TUB4–AID* cells assemble cMTs away from the SPB after nocodazole washout. Time-lapse sequences of *TUB4–AID yeGFP–TUB1 SPC42–mCherry* cells. After nocodazole washout in the presence of IAA and α-factor (as outlined in *Figure 7a*) cells were placed on a MatTek dish and imaged immediately. Images were acquired every 5 min and post-processed with maximum intensity projection.
DOI: https://doi.org/10.7554/eLife.39932.014

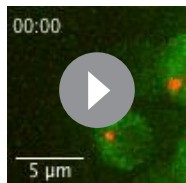

**Video 2.** *TUB4–AID* cells assemble cMTs away from the SPB after nocodazole washout. Time-lapse sequences of *TUB4–AID yeGFP–TUB1 SPC42–mCherry* cells with nocodazole washout as described in Video 1.
DOI: https://doi.org/10.7554/eLife.39932.015

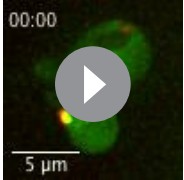

**Video 3.** cMTs assemble at the SPB of WT *TUB4* cells after nocodazole washout. *TUB4 yeGFP–TUB1 SPC42–mCherry* cells were treated and imaged as described in *Videos 1* .
DOI: https://doi.org/10.7554/eLife.39932.016

the schmoo tip (*Figure 7e and f*). By contrast, most *stu2^TOGAA^* and *stu2^ΔCC^* cells were impaired in cMT formation (*Figure 7e*). Only about 20% of *stu2^TOGAA^* and *stu2^ΔCC^* cells re-assembled cMTs, which were shorter and much less robust than cMTs in *STU2–AID STU2* cells (*Figure 7e and f*). About half of the *stu2^ΔMT^* and *stu2^ΔC^* cells assembled cMTs that were shorter than those of *STU2* WT cells. The other half of *stu2^ΔMT^* and *stu2^ΔC^* cells was devoid of cMTs (*Figure 7e*, quantification). Whenever *STU2* mutant cells assembled cMTs, these were associated with the SPB (*Figure 7e*, top). Thus, the TOG domains and the dimerizing coiled-coil domain of Stu2 were most important for cMT formation in the presence of γ-TuSC. Stu2^MT^ had an important impact on cMT formation.

Taken together, our experiments indicate two in vivo functions for Stu2. (1) Stu2 is important for stabilization of the γ-TuSC oligomer at the minus end of detached cMTs (*Figure 6*). (2) Stu2 is crucial for the nucleation of cMTs independently of the presence of γ-TuSC (*Figure 7*).

## Discussion

The functions of TOG domain proteins at the MT minus end could be structural or enzymatic, of general importance or context-specific. The ability to measure the binding affinities and MT nucleation activity of purified Spc72, Stu2 and γ-TuSC in vitro, in combination with the analysis of the function of these proteins in vivo, has now shed light on the role of Stu2 and its cooperation with the γ-TuSC during cMT nucleation.

Here, we show that the purified Spc72N–Stu2 complex assembles MTs from α/β-tubulin subunits in the form of asters in the absence of γ-TuSC. This aster-forming activity was strongly dependent on the MT-binding region in Stu2. Mutations in the two TOG domains did not affect MT aster numbers or MT density in the asters, while overall MT nucleation activity was only slightly reduced by these mutations (*Figure 3b,f* and *Figure 4b*). γ-tubulin-independent MT nucleation reactions that involve TOG-domain proteins have been observed before (*Roostalu et al., 2015*; *Woodruff et al., 2017*). We propose that a Stu2 scaffold that is organized by Spc72N stabilizes a first α/β-tubulin seed through its MT-binding site, in accordance with the strict requirement for this domain in Spc72N–Stu2-dependent MT nucleation (*Figure 3b*). Dimerization of Stu2 is probably important for this nucleation reaction as the Spc72N–Stu2^ΔCC^ complex lacked MT nucleation activity (*Figure 3*).

γ-TuSC-independent MT nucleation by Stu2 from unattached kinetochores has been reported (*Gandhi et al., 2011*). Our data now suggest that nucleation of cMTs in the cytoplasm away from the SPB in Tub4-depleted cells is critically dependent on Stu2. These cMTs lacked Spc72, Spc97 and Spc98 but were associated with Stu2 (*Figure 7*). The absence of Spc72 suggests that another scaffold assembles Stu2 into a MT nucleation competent state for the ectopic cMT nucleation reaction. In WT cells, γ-tubulin-independent cMT nucleation events away from the SPB were rare, so we have to assume that the Spc72–γ-TuSC system at the SPB is dominant over the Stu2-driven ectopic cMT nucleation. This ensures that cMTs are anchored to the SPB with a defined polar orientation. Such an orientation is essential for the function of cMTs in nuclear positioning and for orientation of the mitotic spindle (*Hwang et al., 2003*).

Our in vitro data indicate that Stu2 binds to γ-TuSC (*Figure 2d*) via its MT-binding site (*Wang and Huffaker, 1997*). This is in line with the recent observation that human γ-tubulin binds to the MT-binding site of XMAP215 in vitro (*Thawani et al., 2018*). We suggest that the interaction of the C-terminus of Stu2 with Spc72 (*Usui et al., 2003*), the binding of Stu2^MT^ to γ-tubulin and the

binding of the CM1 of Spc72 to Spc97/Spc98 (*Knop and Schiebel, 1997*) jointly promote the high-affinity interaction of Spc72–Stu2 and γ-TuSC that we have observed in MST measurements. We propose that Stu2 functions structurally as a clamp that joins Spc72 together with γ-TuSC.

Spc72N–Stu2 assembles the γ-TuSC into circular oligomers. Spc72N–Stu2$^{TOGAA}$ showed γ-TuSC oligomerization activity comparable to that of Spc72N–Stu2$^{WT}$, suggesting that the MT polymerization activity of Stu2 is not required for γ-TuSC oligomerization. The addition of γ-TuSC increased the in vitro MT nucleation activity of Spc72N–Stu2 by a factor of about 3. This increase is explained by the ability of the γ-tubulin ring to sustain less favoured lateral interactions of α/β-tubulin subunits. The two TOG domains in Stu2 probably promote the recruitment of the first α/β-tubulin subunits to the γ-TuSC template. As the orientation of Stu2 in such a complex is predetermined by the interaction of its C-terminus with the N-terminally located binding site in Spc72 (*Usui et al., 2003*), each TOG domain orients the α-tubulin subunit of tubulin towards a γ-tubulin molecule in the γ-TuSC oligomer (*Ayaz et al., 2012*). Structural studies are needed to reveal the precise mechanism of the Spc72–Stu2–γ-TuSC cMT assembly machinery.

The in vivo situation is even more complex because at least two proteins, Spc72 and Kar9, recruit Stu2 to the minus end of cMTs. The recruitment role of Kar9 is consistent with the reported interactions of Kar9 with Stu2 and γ-tubulin Tub4 (*Chen et al., 1998*; *Cuschieri et al., 2006*; *Moore and Miller, 2007*; *Usui et al., 2003*) (illustrated in *Figure 8*). Stu2 probably binds to Kar9 in the Tub4–Kar9 complex via its C-terminal 33 aa of Stu2 (*Moore and Miller, 2007*).

Upon removal of both Stu2 recruitment factors (*spc72$^{\Delta Stu2}$* and Kar9 depletion), γ-TuSC dissociated from the minus end of detached cMTs (*Figure 6g*), reflecting the low affinity of the Spc72–γ-TuSC interaction without Stu2 (*Figure 1d* and *Figure 2c*). Our data suggest that cMTs with a detached γ-TuSC cap disassemble, probably because factors that protect the MT minus end in the absence of the γ-tubulin cap in higher eukaryotes, such as the protein CAMSAP2 (*Jiang et al., 2014*; *Wiese and Zheng, 2000*), are not encoded by the yeast genome.

The TOG domains in the Kar9–Stu2 layer are most probably positioned more distantly away from the cMT minus end than those within Spc72–Stu2 (*Figure 8*). The two Stu2 layers likely function in a sequential order: Spc72–Stu2 functions to add the first α/β-tubulin subunits and then Kar9–Stu2 adds the next α/β-tubulin molecules (*Figure 8*). A single XMAP215 molecule that is bound via its MT-binding site to γ-tubulin has, with five TOG domains, a greater reach to add α/β-tubulin subunits to the growing MT than a single Stu2 layer with only two TOG domains (*Widlund et al., 2011*). Binding of Stu2 to Spc72 and Kar9 probably compensates for the smaller number of TOG domains in Stu2.

The ratio of Spc72:Tub4:Spc97:Spc98 molecules at the cMT minus end is comparable to that of Spc110:Tub4:Spc97:Spc98 at the minus end of nuclear MTs (*Erlemann et al., 2012*; *Kollman et al., 2010*). Green fluorescent protein (GFP) counting suggests that 70 to 90 Stu2 molecules are associated with the cMT minus end. Thus, the molar Stu2:Tub4 ratio is at least 5:1 at a cMT nucleation site (*Erlemann et al., 2012*; *Kollman et al., 2010*). By contrast, Stu2 does not bind to the nuclear Spc110 (*Usui et al., 2003*) and Stu2 depletion did not strongly impact the nucleation of nMTs (*Figure 7*). Furthermore, Kar9 is a cytoplasmic protein without a nuclear function (*Miller and Rose, 1998*). Whether Stu2 binds directly to a nuclear γ-TuSC ring via its MT-binding site remains to be analysed. In any case, the much higher affinity of Spc110 for γ-TuSC makes a clamp-like function of Stu2 unnecessary. Thus, the functions of Stu2 at MT nucleation sites are at least in part context-specific.

In summary, γ-TuSC assembles into a functional cMT nucleation site with the assistance of the Spc72–Stu2 complex (*Figure 8*). The CM1 of Spc72 and the binding of Stu2 to Spc72 are important for this γ-TuSC oligomerization. After the oligomerization of Spc72–Stu2–γ-TuSC, the TOG domains in Stu2 probably assist in the oriented recruitment of the first α/β-tubulin subunits to the γ-TuSC (*Figure 8a*, right). A second α/β-tubulin recruitment phase is organized by the Kar9–Stu2 interface (*Figure 8a*, left). This γ-TuSC-dependent cMT nucleation system at the SPB is dominant over an alternative cMT assembly pathway that we observed upon Tub4 depletion (*Figure 8b*). The nucleation of nMT by Spc110 and γ-TuSC is not Stu2-dependent, or is Stu2-dependent to a lesser degree, suggesting that the role of Stu2 in MT nucleation is context-specific.

TOG domain proteins function in MT nucleation in a number of organisms (*Chen et al., 1998*; *Flor-Parra et al., 2018*; *Thawani et al., 2018*; *Usui et al., 2003*). Binding of γ-tubulin to the MT-binding site of TOG-domain proteins could be a conserved principal (*Thawani et al., 2018*)

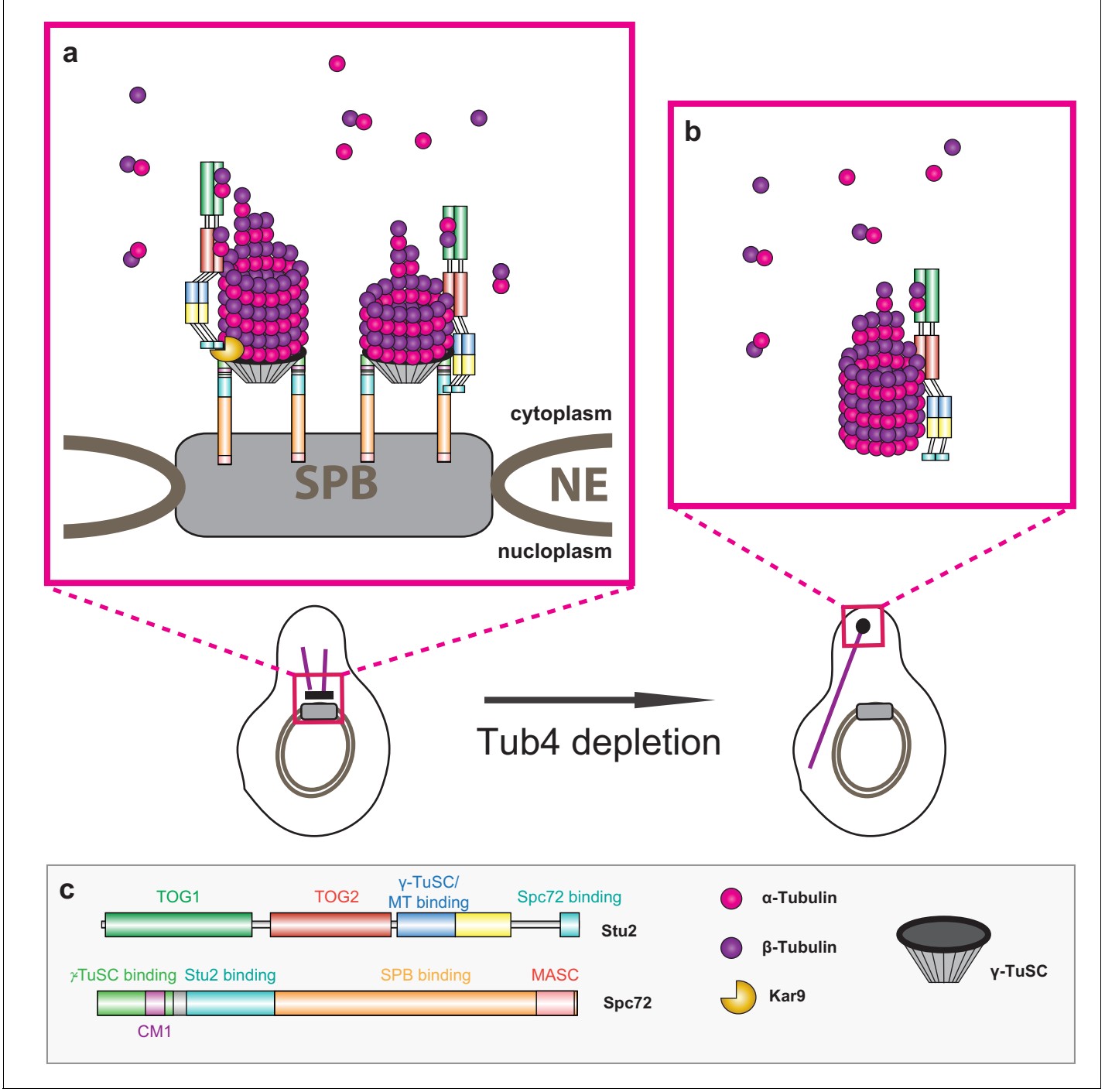

**Figure 8.** Model of Stu2-assisted cMT nucleation. (**a**) Right: initiation of cMTs assembly through stabilisation of the Spc72–γ-TuSC interaction and the oriented recruitment of the first α/β-tubulin molecules. Left: further cMT growth promoted by the Kar9–Stu2 interaction. (**b**) In the case of Tub4 depletion, Stu2 can promote the assembly of cMTs away from the SPB. (**c**) Colour code for the domain organisation of Stu2 and Spc72 and legend for all of the proteins and complexes shown.

DOI: https://doi.org/10.7554/eLife.39932.017

(*Figure 2d*). Additional studies are needed to reveal when TOG-domain proteins require γ-tubulin complexes for MT nucleation or assemble MTs without them.

## Materials and methods

### Yeast methods

Most yeast strains that were used in this study are derivatives of ESM356-1 (MATa *ura3-52 trp1Δ63 his3Δ200 leu2Δ1*) and are listed in *Supplementary file 1*. *Supplementary file 2* lists the plasmids used in this study. Genes were deleted or tagged endogenously using PCR-based methods (*Janke et al., 2004*; *Knop et al., 1999*). Tagging was verified by PCR, microscopy and immunoblotting. Plasmids with the backbone of pRS304, pRS305 and pRS306 (*Sikorski and Hieter, 1989*) were linearized and stably integrated into the yeast genome by homologous recombination. For the $G_1$ arrest, cells were treated with 10 µg/ml α-factor. For nocodazole arrest in SC-complete medium, cells were incubated with 1% peptone and 15 µg/ml nocodazole. For auxin-depletion experiments, proteins were tagged with 3HA-IAA and depleted with an IAA (3-Indoleacetic acid, Sigma-Aldrich I2886) concentration of 1 mM at 30°C. For drop tests, cells derived from S288C (YPH499; *Sikorski and Hieter, 1989*) were serially diluted 1:10 and spotted on either SC-Leu or 5-FOA plates. Plates were incubated at 30°C for 2 d or 5 d. We used alkaline lysis and TCA to prepare protein extracts from yeast cells (*Knop et al., 1999*).

### Antibodies

Antibodies were mouse anti-GFP (1:1000; Roche 11814460001), rabbit anti-GFP (1:2000; homemade rabbit BS 23.4.08), goat anti-GST (1:2000; GE Healthcare 27-4577-01), mouse anti-HA (1:50; 12CA5), rabbit anti-Spc97 (1:500; homemade rabbit AT), rabbit anti-Stu2 (1:200; homemade), goat anti-Tub4 (1:250; homemade), rabbit anti-Spc72 (1:300; homemade) and mouse anti-tubulin (1:100; homemade WA3 antibodies). Secondary antibodies were donkey anti-rabbit IRDye 800CW (1:10,000; LI-COR Product-number: 926–32213), donkey anti-goat A680 (1:10,000; Invitrogen, A21084), donkey anti-mouse A680 (1:10,000; Invitrogen, A10038), goat anti-mouse HRP (1:10,000; Jackson 115-035-068), rabbit anti-goat HRP (1:5,000; Jackson 305-035-045), donkey anti-rabbit HRP (1:5,000; Jackson 711-035-152) and goat anti-rabbit HRP (1:10,000; Jackson 111-035-045).

### Microscopy

Microscopy was performed in sterile filtered synthetic complete (SC) medium at 30°C. Log-phase cells were treated as annotated and immobilized on a glass bottom dish (P35G-1.5–14C; MatTek Corporation). In advance, the dish was coated with 100 µl of ConA (6% Concanavalin A type IV, 25 mM Tris-HCl pH 7, 1 mM MnCl$_2$) for 2 min and then washed twice with water. Cells were allowed to attach for 5 to 15 min, subsequently washed with water, and finally SC medium was added. For imaging, a DeltaVision RT system (Olympus IX71 based; Applied Precision Ltd.) equipped with a Photometrics CoolSnap HQ camera (Roper Scientific), a 100 × 1.4 NA Super–Plan Apochromat oil objective (Olympus), a four-color Standard InsightSSI module light source including a laser-based hardware autofocus, and a Workstation with a CentOS operating system and SoftWoRx software (Applied Precision Ltd.) was used. For photobleaching experiments, a 50 mW 488 nm laser system was used on the same microscope. For deconvolution of images, the softWoRx software package of DeltaVision was used. For live-cell imaging, the system was prewarmed to 30°C and imaging was done with binning 2 × 2. Image analysis was performed with ImageJ 1.46 r (National Institutes of Health, USA) and GraphPad Prism (GraphPad Software).

### Line scans

For line scans of detached cMTs of *kar1Δ15* cells both wavelengths were acquired for each z-stack, followed by the next stack. After deconvolution of the images, MTs from single z-planes were analysed in ImageJ by drawing a line along the MT, and signal intensities were measured with the plot profile command. Signal normalization was performed against the highest signal intensity of the measured signal in Excel (Microsoft).

### Measurement of molecule numbers

For measuring molecule numbers of proteins tagged with yeGFP, *kar1Δ15* cells and *CSE4-yeGFP Δste2* cells were α-factor treated and mixed. First, all 21 stacks of the yeGFP wavelength were acquired followed by all stacks of the second fluorescence signal in order to localize the yeGFP

signal on the detached cMT. A reference image of an empty Matek dish was acquired with the imaging buffer and the same acquisition settings to correct for variations of the field illumination in the GFP-channel. After subtracting the reference image from all the GFP-stacks, the brightest signal intensity from single z-planes of the 21 stacks was analysed in ImageJ and corrected for the background close to the measured signal as published (*Erlemann et al., 2012*).

## Fluorescence recovery after photobleaching (FRAP) experiments

For FRAP experiments examining *kar1Δ15* cells with a yeGFP-tag on the protein of interest, the yeGFP signal was first focused in the z-plane. Three pre-bleach images were taken, followed by 50 images after photobleaching, for an expected signal recovery halftime of 1 s. Photobleaching was performed with a 100x objective and a 488 nm laser with the power of 12% for Stu2 and 17% for Spc72 for 50 msec. Normalization of the FRAP data was performed as described by *Erlemann et al. (2012)* with the GraphPad Prism plotting function with some changes. Fluorescence intensities of the yeGFP-tagged proteins were measured and normalized using the Phair double-normalization method (*Phair et al., 2004*), setting the fluorescence intensity at the moment of bleaching to 0. A curve reflecting the total mean of the fluorescence recovery was corrected for overall bleaching by measuring the fluorescence intensity decrease of a neighbouring cell with the formula $y = (y_0 - Plateau) * e^{(-Kd*x)} + Plateau$ (One-phase decay in GraphPad Prism). The constant $K_d$ was extracted from the control measurement and the recovery curve was calculated with $y = y_0 + (Plateau - y_0) * (1 - e^{-(K+Kd)x})$ (as in One-phase association in GraphPad Prism). The half-time was extracted from that curve with $\ln(2)/K$.

## γ-TuSC purification

The γ-TuSC was purified as described by *Gombos et al. (2013)*.

## Purification of 6His-Stu2

The purification protocol for 6His-Stu2 and its mutant versions is a variation of the published protocols (*Al-Bassam et al., 2006*; *van Breugel et al., 2003*). 6His-Stu2 proteins were expressed in SF21 insect cells after infection for 48 hr with 1% baculovirus. Cells were harvested by centrifugation at 2000 rpm for 10 min and stored at −80°C. The pellet from 800 ml culture was resuspended in HB200 (50 mM Hepes, pH 7.5, 200 mM KCl, 1 mM MgCl$_2$, 5% glycerol[(w/v]) together with one tablet of Protease Inhibitor Cocktail (Roche, 11873580001) and 1 mM DTT. Cells were lysed on ice with a Polytron (Polytron PT 3100 dispersion unit, Kinematica) for 5 min at 4000 rpm. The lysate was incubated with 0.25% Brij35 for 10 min at 4°C and centrifuged at 4°C for 30 min at 15,000 rpm in a SS-34 rotor. The supernatant was bound to 500 mg equilibrated Ni-beads (Protino Ni-TED; Machery-Nagel) for 2 hr at 4°C. Beads were washed four times with HB200, and proteins were eluted in 500 mM imidazole. The eluate was injected into an ÄKTA system (GE Healthcare), which was equilibrated with HB200 and purified through size exclusion chromatography (HiLoad 16/60 Superdex 200 column, GE Healthcare) at 4°C. The fractions were analysed by SDS-PAGE and those containing Stu2 were pooled, concentrated and stored at −80°C. The bar graphs were designed with DOG1.0 (*Ren et al., 2009*).

## Purification of GST-Spc72N

The purification of GST–Spc72N followed a variation of the protocol published by *Lin et al. (2014)*. A preculture of *Escherichia coli* BL21-CodonPlus was incubated at 37°C in LB-Amp-Cam (100 µg/ml Ampicillin, 30 µg/ml Chloramphenicol) overnight, diluted to OD$_{600}$ 0.2 in LB-Amp-Cam and incubated at 22°C for 1 hr at 100 rpm. Induction was carried with 200 µM IPTG for 20 hr at 22°C. Cell pellets of 1 l culture were harvested by centrifugation at 3000 g for 15 min and stored at −80°C. The pellet was resuspended in lysis buffer (40 mM K-Hepes pH 7.5, 250 mM KCl, 10 mM EDTA, 1 mM EGTA, 10% glycerol [w/v], 4 mM DTT) together with one tablet of Protease Inhibitor Cocktail and 1 mM PMSF. Cells were lysed on ice by sonication (Bandelin Sonopuls GM 200 equipped with a Bandelin electronic UW200) with a KE76 tip five times for 30 s with 50% power. The lysate was centrifuged for 30 min at 4°C at 26,890 g in a SS-34 rotor. The supernatant was incubated with 750 µl equilibrated GST-beads (Protino Gluathione Agarose 4B; Machery-Nagel) for 2 hr at 4°C. The beads were washed three times with lysis buffer and the protein was eluted with 30 mM L-glutathione in

lysis buffer. The eluate was injected into an ÄKTA system (GE Healthcare) and purified through size-exclusion chromatography (HiLoad 16/60 Superdex 200 column, GE Healthcare) at 4°C. The chromatography was done with TB150 (50 mM Tris-HCl pH7.0, 150 mM KCl, 1 mM MgCl$_2$, 1 mM EDTA). The fractions were analysed by SDS-PAGE and those containing Spc72 were pooled, concentrated and stored with 10% glycerol at −80°C.

### Co-purification of Spc72N–Stu2

An *E. coli* pellet of 1 l from GST–Spc72N-expressing cells was resuspended in HB200 together with one tablet of Protease Inhibitor Cocktail and 1 mM DTT. Lysis and centrifugation were carried out as for Spc72 purification. The supernatant containing GST–Spc72N was bound to equilibrated GST-beads for 2.5 hr at 4°C. Meanwhile, a pellet of 800 ml SF21 insect cells expressing 6His–Stu2 was resuspended in HB200 together with one tablet of Protease Inhibitor Cocktail and 1 mM DTT. Cells were lysed with the Polytron, 0.25% Brij35 was added for 10 min and the lysate was centrifuged as mentioned in Stu2 purification. The GST–Spc72N beads were washed five times with HB200 and the 6His–Stu2 lysate was added after dilution with HB200 containing 0.125% Brij35. After incubation overnight at 4°C, the beads were washed five times with HB200 containing 0.125% Brij35 and proteins were eluted with 50 mM L-glutathione in HB200 with 0.125% Brij35. The elutions were loaded on an ÄKTA system (GE Healthcare) with an HiLoad 16/60 Superdex 200 column (GE Healthcare), which was equilibrated with HB200 at 4°C. Gel filtration fractions were analysed by SDS-PAGE and those containing GST–Spc72N and 6His–Stu2 were pooled. The purified complexes were stored at −80°C with 0.125% Brij and 5% glycerol.

### Purification of Spc110N

Purification of the N-terminal version of Spc110N (GST–Spc110N$^{1-220}$) and its mutant version Spc110N$^{CM1}$ (GST–Spc110$^{1-220-QA}$) were performed as described by *Lin et al. (2014)*.

### Pulldown by GST-tagged proteins

The pulldown protocol is a variation of a protocol published by *Lin et al. (2014)*. Recombinant proteins (see purification of proteins) were pre-cleared by centrifugation at 20,817 g at 4°C. 6His–Stu2 and γ-TuSC proteins were diluted with HB100 (40 mM Hepes pH 7.5, 100 mM KCl, 1 mM EGTA, 1 mM MgCl$_2$) and incubated with GST-beads for 30 min on ice. The supernatant was mixed with GST or GST–Spc72N to a final concentration of 2 µM for Stu2, 150 nM for γ-TuSC, 1 µM for GST and 1 µM for GST–Spc72N. The GST–Stu2$^{TOG}$ (aa 1–590 of Stu2) and GST–Stu2$^{MT}$ (aa 451–684 of Stu2) were purified as described previously (*Widlund et al., 2012*). Pig brain tubulin and γ-TuSC were diluted with HB100 and incubated with glutathion beads for 30 min on ice. The supernatant was mixed with GST, GST–Stu2$^{TOG}$ or GST–Stu2$^{MT}$ to a final concentration of 500 nM for tubulin or γ-TuSC and 1 µM for GST, GST–Stu2$^{TOG}$ or GST–Stu2$^{MT}$. After an incubation for 30 min, the protein mixture was incubated at 4°C for 1 hr with equilibrated glutathion beads. Beads were washed once with washing buffer (pulldown buffer with 0.1% Nonidet P40; Biochemica) and transferred to a new tube. Beads were washed six times and the remaining buffer was removed with a squeezed pipette tip. Samples were incubated with sample buffer at 65°C for 5 min and analysed by immunoblotting.

### Pulldown of tubulin with 6His–Stu2

Pig brain tubulin was diluted to 30 µM together with 30 µM nocodazole in BRB80 (80 mM PIPES, pH 6.8, 1 mM MgCl$_2$, 1 mM EGTA), precleared by centrifugation in a S100AT3 rotor at 4°C for 5 min at 353,000 g, and half of the supernatant was incubated with Ni-beads (Qiagen) equilibrated with HB300 (50 mM K-Hepes, pH 7.5, 300 mM NaCl, 1 mM MgCl$_2$, 5% glycerol) for 30 min at 4°C. The tubulin supernatant was taken without transferring any beads. 2 µM of Stu2 proteins (which were pre-cleared by centrifugation at 20,817 g at 4°C) were incubated together with 2 µM tubulin from the supernatant on equilibrated Ni-beads in HB300 for 1 hr at 4°C. Beads were washed once with HB300 supplemented with 0.5% Nonidet P40 (Biochemica) and 20 mM imidazole, and transferred to a new tube. Then beads were washed two more times and the remaining buffer was removed with a squeezed pipette tip. Beads were treated with sample buffer at 65°C for 5 min and the supernatant was analysed by immunoblotting.

## Gel filtration of 6His–Stu2 with tubulin

Pig brain tubulin was diluted with BRB80 12.5% glycerol (w/v), centrifuged at 4°C for 5 min at 353,000 g in a S100AT3 rotor, and half of the supernatant was used for the experiments. Stu2 (which was pre-cleared by centrifugation at 20,817 g at 4°C) and tubulin were diluted together to 5 µM in HB100 supplemented with 10% glycerol and incubated for 40 min on ice. Size-exclusion chromatography was performed with a Superose six column (10/300 GL, GE Healthcare) on an ÄKTA system (GE Healthcare). Fractions were treated with sample buffer and analysed by immunoblotting.

## In vitro MT nucleation assay

For in vitro nucleation of MTs, unlabelled and Cy3-labelled (following protocol by *Hyman et al. (1991)*) pig brain tubulin was melted by hand, diluted to 40 µM in BRB80 with 12.5% glycerol and centrifuged in a S100AT3 rotor at 4°C for 5 min at 353,000 g. The upper half of the supernatant was adjusted to 24 µM. Proteins were mixed and incubated for 1 hr in BRB80 with 12.5% glycerol and 1 mM GTP on ice. 2.5 µl of protein mixture was mixed with 2.5 µl tubulin (final tubulin concentration 12 µM) and incubated for 10 min on ice. The following concentrations were used: 50 nM for γ-TuSC and 100 nM for Stu2, Spc72N or the purified Spc72N–Stu2 complexes. Each assay was incubated at 37°C for 16 min and 20 µl of 1% glutaraldehyde was added with a cut pipette tip. After 7 min 30 s incubation at 23°C, 400 µl of cold BRB80 was added and the tube was inverted for mixing. Coated coverslips (15 min of Poly-L-lysine [Sigma-Aldrich P8920]) were added on top of adaptors for Corex tubes (15 ml, No 8441) and supplied with 5 ml of BRB80 in 10% glycerol. An assay volume of 50 µl was added just underneath the liquid cushion and spun at 25,530 g, at 4°C for 1 hr in a HB6 rotor. Coverslips were fixed with ice-cold methanol and washed twice with cold PBS. Drops of Citifluor fixed the coverslips on objective slides, which were finally sealed with nail polish. Twenty images were acquired for the asters with a 63x objective and for the MTs with a 100x objective on a Zeiss Axiovert 200M equipped with a Cascade 1 k EMCCD camera (Cascade Photometrics, Roper Scientific, Inc.) and with a 63 × 1.4 NA Plan Apochromat oil objective (Zeiss) and a 100 × 1.46 NA Plan Apochromat oil objective (Zeiss), a HBO100 lamp and a Ludl MAC 5000 (Ludl Electronic Products Ltd.) with FWSHCDC controlling a DIC/488/555/647 filter wheel. The system was managed with the software VisiView (Version 2.1.4, Visitron Systems GmbH). Analysis was performed with ImageJ and the aster number was corrected by the mean number of nucleated MTs in 20 fields of the tubulin-only control taken with a 100x objective. The MT formation activity was derived by a measurement of the mean intensity of the maximum projection of 20 acquired images. The tubulin-only control intensity was subtracted from other values to correct for the background.

## In vitro reconstitution of γ-TuSC–Spc72N and γ-TuSC–Spc72N–Stu2 oligomers

Purified γ-TuSC was mixed with GST–Spc72N$^{CM1}$ or GST–Spc110N complex in 1:4 molar ratio (4.6 µM:18.6 µM) in HB100-50 buffer (50 mM Hepes, pH7.5, 100 mM KCl, 1 mM EGTA, 1 mM MgCl$_2$) with salt as indicated in the figure legends. After incubating for 15 min at 4°C, the protein mixture was applied to a gel filtration column (Superose 6 PC 3.2/30, GE Healthcare, UK). Elution profiles were recorded at absorbance at 280 nm. Proteins were detected by Coomassie Blue or immunoblotting. For *Figure 2e* and *Figure 2—figure supplement 2d*, purified γ-TuSC was mixed with GST–Spc72N, GST–Spc72N–6His–Stu2$^{WT}$ or GST–Spc72N–6His–Stu2$^{TOGAA}$ to a final concentration of 4.6 µM for γ-TuSC and 2.3 µM for the other proteins in HB100 buffer with 10% glycerol. Proteins were incubated on ice for 40 min, centrifuged at 20,817 g for 10 min at 4°C and 50 µl were loaded on a gel filtration column (Superose 6 PC 3.2/30) equilibrated with HB100. Proteins were detected by immunoblotting. Electron microscopy analysis was performed as described by *Lin et al., 2016*. A High Molecular Weight (HMW) Kit (GE Healthcare, 28-4038-42) was used for calibration of the Superose 6 PC 3.2/30 column.

## Binding-affinity measurement by microscale thermophoresis (MST)

MST assays were carried out with a Monolith NT.115 instrument (NanoTemper, Munich, Germany). Each titration curve consisted of 16 points prepared from a serial dilution of analytes and a constant concentration of the fluorescein-labelled ligand. Purified γ-TuSC and Stu2 were fluorescently labelled with GREEN-NHS (NanoTemper, Munich, Germany). To measure the affinity between γ-TuSC and its

binding partners, 2X serial diluted recombinant Spc110N, Spc72N, Stu2 or Spc72N-Stu2 complexes were titrated against 50 nM of labelled γ-TuSC. To measure the binding affinity between Stu2 and Spc72N, increasing concentrations of recombinant Spc72N were titrated against 150 nM of labelled Stu2. Experiments were carried out in HB100-50 buffer supplemented with 0.05% (w/v) Tween-20. The samples were loaded into standard glass capillaries (Monolith NT capillaries, NanoTemper, Munich, Germany). MST assays were performed with 60% and 80% LED power using a green filter and 40%, 60% and 80% MST power. The normalized fluorescence readings (Thermophoresis + T jump or T-jump only) were plotted as a function of analyte concentration and the curve fitting and dissociation constant $K_d$ calculation were performed with Nanotemper software. In all cases, MST data were fitted on the based of a single-site model.

### Statistical analysis

Statistical analysis of the data was carried out with GraphPad Prism and performed as two-tailed unpaired t-tests with a confidence interval of 95%. For MST measurements, the statistical analysis was done in Excel. The number of repeated experiments is indicated in the figure legend together with the sample size and the P-values. In the MST measurements, when the data points of a single concentration value were deviated from normal intensity range, they were considered as outliers and excluded from the analysis. Otherwise no data points were excluded. Experiments are technical replicates if not stated differently. The experiments were not randomized and the investigators were not blinded to allocation during the experiments and outcome assessment.

## Acknowledgements

The authors thank Dr. M Mayer (ZMBH) for the use of the MST equipment. JG is a member of the Hartmut Hoffmann-Berling International Graduate School.

## Additional information

### Funding

| Funder | Grant reference number | Author |
| --- | --- | --- |
| Deutsche Forschungsge-meinschaft | Schi295/4-3 | Judith Gunzelmann Tien-chen Lin Annett Neuner Ursula Jäkle |

The funders had no role in study design, data collection and interpretation, or the decision to submit the work for publication.

### Author contributions

Judith Gunzelmann, Conceptualization, Formal analysis, Investigation, Writing—original draft; Diana Rüthnick, Conceptualization, Formal analysis, Investigation; Tien-chen Lin, Wanlu Zhang, Formal analysis, Investigation; Annett Neuner, Ursula Jäkle, Investigation; Elmar Schiebel, Conceptualization, Supervision, Funding acquisition, Validation, Writing—original draft, Writing—review and editing

### Author ORCIDs

Diana Rüthnick (iD) http://orcid.org/0000-0001-6365-4050
Elmar Schiebel (iD) http://orcid.org/0000-0002-3683-247X

### Decision letter and Author response

Decision letter https://doi.org/10.7554/eLife.39932.022
Author response https://doi.org/10.7554/eLife.39932.023

## Additional files

### Supplementary files
• Supplementary file 1. Yeast strains used for this study.
DOI: https://doi.org/10.7554/eLife.39932.018
• Supplementary file 2. Plasmids used for this study.
DOI: https://doi.org/10.7554/eLife.39932.019
• Transparent reporting form
DOI: https://doi.org/10.7554/eLife.39932.020

### Data availability
All data generated or analysed during this study are included in the manuscript and supporting files.

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
