## [Decision Letter]

[Editors’ note: a previous version of this study was rejected after peer review, but the authors submitted for reconsideration. The first decision letter after peer review is shown below.]

Thank you for submitting your work entitled "TOG domain independent microtubule nucleation activity of Stu2 protein at centrosomes" for consideration by *eLife*. Your article has been reviewed by a Senior Editor, a Reviewing Editor, and three reviewers. The reviewers have opted to remain anonymous.

Our decision has been reached after consultation between the reviewers. Based on these discussions and the individual reviews below, we regret to inform you that your work will not be considered further for publication in *eLife*.

As you will see from the comments below all three reviewers had significant concerns about controls, methods, and interpretations for both the biochemical and in vivo experiments. Together, this led them to a decision that this paper should be rejected.

*Reviewer #1:*

Gunzelmann et al., present a study on Stu2, a member of the TOG domain-containing microtubule (MT) polymerase family. Stu2 is well characterized as acting at MT plus ends. In this work the authors explore the role of Stu2 at the centrosome/ yeast spindle pole body (SPB), where Stu2 localizes in yeast cells. On the nuclear side of the nuclear membrane Spc110 serves as the Ɣ-TuSC receptor, while on the cytoplasmic face Spc72 is the Ɣ-TuSC receptor. This work presents the novel finding that Stu2 facilitates Spc72 binding to Ɣ-TuSC and possibly promotes Ɣ-TuSC oligomerization in vitro. The authors then look at Stu2 localization in vivo using mutants with detached MT minus ends. They find that Stu2 localizes to MT minus ends in the absence of the Stu2 binding site in Spc72 and that Kar9 plays a role in this recruitment. The authors then present additional in vitro experiments to look at microtubule aster formation. They suggest that Stu2 can promote MT aster formation in the absence of functional TOG domains in the presence of Spc72.

While this work contains several interesting observations, many experiments are lacking important controls to support the conclusions made. In addition, in its current form I do not see that it dramatically advances our understanding of TOG domain-containing proteins nor is it of sufficient quality for *eLife*. I recommend that the authors revise their work based on the following comments to submit elsewhere.

Essential revisions:

1) Does Stu2 directly interact with Ɣ-TuSC? This should be addressed.

2) The gel filtration experiment in Figure 2D is missing standards and the void volume is not labeled. How many times was this experiment repeated? This should be listed in the legend. What concentrations of proteins were used?

3) Ɣ-TuSC oligomerization appears to be enhanced in the presence of Spc72N-Stu2TOGAA (defective in tubulin binding). This experiment should also be done with Spc72 and wildtype Stu2.

3) The authors should quantify Stu2 fluorescence at the cMT minus end in the spc72Δstu2/Kar9-depletion conditions. Is Stu2 completely absent or are factors beyond Spc72/Kar9 also recruiting Stu2 to the minus end?

4) As the authors currently present it, the mechanism of Stu2-dependent MT assembly is unclear. The experiments in Figures 6 and 7 are difficult to interpret because controls are lacking. For example, the Stu2 mutants alone were not analyzed in Figure 6. Is there any affect of Stu2 or the Stu2 mutants in the presence of Ɣ-TuSC, but in the absence of Spc72 (Figure 7)?

5) The concentrations of tubulin, and all other proteins used in Figures 6 and 7 are not listed in the methods or legends. This is critical to include. Also, were different concentrations tested? How were the concentrations that were used chosen?

6) It is unclear what distinct roles the authors propose for the MT-binding and TOG-binding domains of Stu2 to have in enhanced MT nucleation at the centrosome/ SPB. Are cells with Stu2 MT-binding and TOGAA mutations viable?

7) The authors reference two "nucleation activities" of Stu2, but then posit that the TOG domains may be required for extending Ɣ-TuSC-nucleated seeds. This seems contradictory.

8) The manuscript could benefit greatly from additional editing for clarity and conciseness. There are many grammatical errors as well. The Figure Legends need to be improved for clarity and completeness.

*Reviewer #2:*

This is a review of the article titled "TOG domain independent microtubule nucleation activity of Stu2 protein at centrosomes" by Gunzelmann et al., submitted to *eLife*. The Dis1/Stu2/XMAP215 family of microtubule polymerases are complex, multi-domain proteins that play roles at microtubule plus ends as well as at microtubule organizing centers (MTOCs). While the family's conserved N-terminal tubulin-binding TOG domains play critical roles in microtubule polymerization, less is known about the C-terminal regions that engage MTOC components. Yeast members have significantly diverged from non-yeast members across their C-terminal region. In this manuscript, the authors focus on elucidating the role of Stu2 (the *S. cerevisiae* XMAP215 member) on microtubule nucleation and stability at the spindle pole body (SPB), and how it uses specific determinants to engage other factors and drive microtubule formation/stabilization. The authors show that Stu2 potentiates the interaction between Spc72 and γ-TuSC and in turn promotes γ-TuSC oligomerization which serves to template and nucleate microtubules. While Spc72 recruits Stu2 to the microtubule minus end, the authors identify Kar9 as another factor capable of recruiting Stu2 to the microtubule minus end. The authors probe the role of the Spc72-Stu2 and Kar9-Stu2 interactions on spindle morphology and microtubule stability and length in vivo and find that both sets of interactions play key roles. The authors then analyze the in vitro effects of Spc72 and Stu2 on microtubule nucleation in vitro with and without addition of the γ-TuSC complex and map key regions within Stu2 that are required for microtubule nucleation. The manuscript adds to our understanding of Stu2 and its role in yeast mitosis using a nice blend of in vitro and in vivo analyses. In its current state it lacks some critical experiments that, if present, would best unite disparate pieces in the paper, and help the authors arrive at some of their conclusions.

1) Gel filtration is used to determine whether Spc72 (plus or minus Stu2) causes oligomerization of γ-TuSC, as does Spc110. The results show that Spc72 does shift γ-TuSC to the void volume, and that Stu2 enhances this shift. However, Stu2 also causes a new peak to arise at fraction A11 that could be accounted for by Stu2-mediated dimerization of a γ-TuSC-Spc72 complex. This then calls into question the increase of the γ-TuSC fraction in the void volume: is this really oligomerization as seen with Spc110 which generated γ-TuSC filaments, or is it due to cross-linking mediated by dimers of Spc72 and dimers of Stu2? To resolve this, the authors should determine if the Spc72-Stu2 complex does drive γ-TuSC oligomerization – this could be achieved using negative stain EM analysis, similar to work done by Kollman et al., 2010, DIC microscopy, or fluorescence microscopy.

2) Figure 7, the authors should conduct a Stu2 WT +/- γ-TuSC control to best determine how much Spc72-Stu2 augments the MT nucleation activity in vitro.

3) The authors delineated independent roles for Spc72 and Kar9 on Stu2 recruitment to the MT minus end, analyzed the effects of these respective complexes on cMT nucleation, number, and length in cells. However, when the authors then move to in vitro analyses, they concentrated exclusively on the Spc72-Stu2 interaction (plus and minus γ-TuSC) and its effects on microtubule nucleation in vitro. To be comprehensive, it would greatly strengthen the paper to conduct a comparative analysis of the effects of Kar9 and Kar9-Stu2 (plus and minus γ-TuSC) on MT nucleation in vitro with that of Spc72-Stu2.

*Reviewer #3:*

Review of Gunzelmann et al., TOG domain independent microtubule nucleation activity….

This paper describes a novel nucleation activity within the spindle pole component, Spc72 that recruits Stu2 (TOG domain protein) to the cytoplasmic face of the spindle pole body, where it increases MT density and plays a role in nucleating MT asters.

The binding data in Figure 1 and Figure 2 was very convincing. The data in Figure 3 were problematic. Firstly, they state that incubation in a kar1 mutant results in detached cMTS (Pereira et al., 1993; Erlemann, 2012). I looked at both of these papers and was unable to find the percent of detached vs attached cytoplasmic microtubules in the mutant. It is important to quantitate what fraction of the microtubules are detached to differentiate pole binding from minus-end binding. An alternative interpretation for Figure 3c is that observation of Stu2 in the absence of the stu2 binding site in Spc72 is that these microtubules are not detached from the pole.

The FRAP recovery are nicely done and convincing that there are multiple binding sites. It is interesting that in this and panel b the mt minus ends seem to be in the shmoo tip. Since there are known plus-end binding complexes in the shmoo tip, is there concern that the co-localization reflects the juxtaposition of the minus-end to the shmoo tip. This is easily addressed by quantitating the fraction of spc72 or stu2 spots in the tip vs. the presence or absence of mt minus ends.

Figure 4D is a critical panel for interpretation of aforementioned experiments. It is stated that a-g are kar1Δ15. Approximately 50% of the cMT are attached in these mutants. Thus, the reviewer needs to redraw Figure 3a to reflect the fact that 50% are detached and 50% are released. In addition, the reviewer needs to include the attachment/detachment state of the microtubules for the co-localization. Are they only examining detached microtubules, if so, what is the metric for the data in Figure 3? The metric in Figure 4 is clear, mCherry-Tub1, but if 50% are attached, the data without Tub1 become significantly ambiguous.

In summary, this paper documents an important second TOG-independent mechanism in Stu2 for MT formation. Stu2 has critical roles in spindle orientation and kinetochore function, and thus this second site is important to understand. The authors should ensure that for all reports of minus-end binding, that these are indeed detached cytoplasmic microtubules.

[Editors’ note: what now follows is the decision letter after the authors submitted for further consideration.]

Thank you for resubmitting your work entitled "The TOG domain protein Stu2 promotes γ-TuSC oligomerization in cytoplasmic microtubule nucleation" for further consideration at *eLife*. Your revised article has been favorably evaluated by Anna Akhmanova (Senior Editor), a Reviewing Editor, and three reviewers. The manuscript has been improved but there are some remaining issues that need to be addressed before acceptance, as outlined below.

Gunzelmann et al., present a study on Stu2, a member of the TOG domain-containing microtubule (MT) polymerase family. Stu2 is well characterized as acting at MT plus ends. Here the authors explore the role of Stu2 at MT minus ends and its interactions with Ɣ-TuSC. On the nuclear side of the nuclear membrane Spc110 serves as the Ɣ-TuSC receptor, while on the cytoplasmic face Spc72 is the Ɣ-TuSC receptor. Using a combination of in vitro characterization and in vivo work the experiments presented here strongly suggests a role for Stu2 in both the stabilization of the Ɣ-TuSC-Spc72 interaction as well as in assembling detached MTs in the cytoplasm. The study expands our understanding of MT nucleation in yeast. It adds to the field in a unique way and complements the recently published XMAP215/γ-tubulin work from the Petry lab.

However, before the paper can be published, we would like to ask you to address some remaining issues which concern data analysis, paper organization and writing.

Data analysis and interpretation:

1) The authors state that the aster density is lower in Spc72N-Stu2TOGAAA asters than in Spc72N-Stu2 asters. However, the quantification used to determine aster density is 'MT formation activity', or the mean intensity fluorescence of the entire field. That type of measurement encompasses the entirety of MT assembly. Therefore, a better quantification of aster density is required.

Also, it appears that the example images shown in Figure 3E are saturated, making aster density difficult to assess.

2) Is there a suspected reason for the increase in Stu2 fluorescence in spc72Stu2Δ versus Spc72 WT cells (Figure 5E)? If you remove the CM1 region of Spc72 in yeast, is there a decrease in Stu2 fluorescence at the cMT minus end?

3) The authors state that the Stu2 signal in spc72Stu2Δ KAR9-AID +IAA cells is similar to spc72Stu2Δ. The value stated (~0.9) is actually even more similar to the Stu2 fluorescence in Spc72 WT cells (Figure 5E). This quantification, even if it is for a small number of cMTs, should be shown in the figure.

Additionally, it is suggested that the KAR9-AID was not fully degraded under these conditions, and that is why Stu2 is still recruited to the cMT minus end. However, a complete knockdown of KAR9-AID in another yeast strain (Figure 6—figure supplement 1C) is shown. The authors should demonstrate the extent of KAR9-AID degradation in spc72Stu2Δ KAR9-AID +IAA cells. Alternatively, the authors should fluorescently tag the KAR9-AID protein to determine if the few numbers of cMTs that still maintain Stu2 also have Kar9 at the cMT minus ends. A possible interpretation is that there may be an additional factor recruiting Stu2 to cMT minus ends.

4) Comments on the FRAP analysis: The authors claim that there are two pools, a stable and unstable fraction of Stu2 at the minus-ends of microtubules. There is concern that there is a limited pool of soluble Stu2, and the recovery curves (Figure 5F) potentially misleading. The lack of recovery in one sample could be due to limited protein pool (so there is not enough unlabeled protein around to populate the FRAP zone. The authors could report the average fluorescence intensity of the spots (in arbitrary units) to examine whether spots of the same fluorescence are being targeted in each sample.

Improving the organization and writing of the paper:

1) The authors have a lot of interesting data, but because of the complex nature of Stu2's multiple roles as well as the complexity of some of the experiments, a lot of the intricacies are lost upon first reading. It would be specifically helpful to have a section in the introduction that clearly outlines each putative role and interaction of Stu2 that will be tested, and the general strategy used to test each of these roles/interactions.

2) Before each set of experiments, the authors should clearly explain why they are performing the experiment and what they expect to occur (a hypothesis/model). In many areas this information is lacking or vague. For example, subsection “Stu2 stabilizes γ-tubulin at cMT ends” "To further elevate the role of Stu2 at the cMT minus end…". It's unclear what the authors mean by that and it is difficult to follow their line of logic. As much as possible, the authors should reference back to their original models so that the reader can more easily follow.

3) It appears that the authors fit their MST data to a single-site model, but this should be clearly stated in the Materials and methods section.

4) The grammar, primarily in the Introduction, needs some additional work.

5) The authors frequently use the uncommon term: "in dependence of" referring to an effect that is "dependent". Given the wide international readership of *eLife*, we suggest that this phrase be rewritten in a more active form so that people, who do not speak English as a first language, do not confuse the statement with "independent of.."

6) Subsection “Purified Stu2 in complex with Spc72 assembles MT aster” fifth paragraph is confusing and should be rewritten.

7) Discussion section, ninth paragraph, the authors probably mean "ratios" rather than "rations".

8) Figure 7—figure supplement 1D and E are important and should probably be included in a main figure.

9) Figure 8 needs a figure legend and the objects within the figure need to be clearly labeled.

10) Figure 7A, B need labels indicating what is fluorescently tagged.

11) Several times protein constructs or yeast strain acronyms aren't described upon first mention. Upon the first introduction of a new protein construct that is given an abbreviation, a description and the amino acid modifications (if applicable) should be stated.

---

## [Author Response]

[Editors’ note: the author responses to the first round of peer review follow.]

We would like to thank the three reviewers for their helpful and constructive comments. As outlined below, we have addressed all the points that were raised by the reviewers. The manuscript was heavily modified and new data have been added. The title, conclusions and discussion were changed accordingly.

Reviewer #1:

[…]While this work contains several interesting observations, many experiments are lacking important controls to support the conclusions made. In addition, in its current form I do not see that it dramatically advances our understanding of TOG domain-containing proteins nor is it of sufficient quality for eLife. I recommend that the authors revise their work based on the following comments to submit elsewhere.

The reviewer acknowledges that the work contains interesting observations. However, he/she also points out that controls are missing. We followed the suggestion of reviewer 1 and have revised our manuscript addressing the points raised by this reviewer. We also included new data (e.g. new Figure 7) to increase the impact of our manuscript.

This manuscript for the first time compares in vitro and in vivo activities of TOG domain proteins with and without Ɣ-tubulin complexes. Our manuscript shows that Stu2 has a structural role in the oligomerization of Ɣ-TuSC and the attachment of Ɣ-TuSC to the MT minus end. It explains why proteins such as Spc72 with an imperfect CM1 can function in the oligomerization of Ɣ-TuSC and how Stu2 bound to Spc72 facilitates MT nucleation. In addition, we provide evidences that the MT binding site in Stu2 interacts with the Ɣ-TuSC. This probably promotes the assembly of the Spc72-Ɣ-TuSC-Stu2 complex on the cytoplasmic side of the SPB. We further show that the Ɣ-tubulin Tub4 is mainly important to restrict cMT nucleation to the SPB but not for cMT nucleation per se. In contrast, Stu2 is essential for Tub4-dependent and independent cMT nucleation. Taken this together, our manuscript will have a significant impact on the MT nucleation field in general.

Essential revisions:1) Does Stu2 directly interact with Ɣ-TuSC? This should be addressed.

We have addresses this point by MST measurements (Figure 2D) and in vitro binding experiments with Stu2^TOG^ and Stu2^MT^ (Figure 2—figure supplement 1E and F and Figure 2—figure supplement 2F). Indeed, Ɣ-TuSC binds to Stu2 via Stu2^MT^. These data are consistent with the recent publication by Petry and co-workers in Nat Cell Biol indicating that human Ɣ-tubulin interacts with the MT binding site of XMAP215 in gel filtration experiments.

2) The gel filtration experiment in Figure 2D is missing standards and the void volume is not labeled. How many times was this experiment repeated? This should be listed in the legend. What concentrations of proteins were used?

We have added the size markers to Figure 2D (now Figure 2E) (and Dextan blue to other runs to indicate the void). Dextran blue marks the void volume. As outlined in the figure legend to Figure 2D, we show one representative experiment of three independent experiments. All of them showed a much higher oligomerization activity for Ɣ-TuSC + Spc72N-Stu2 comapred to Ɣ-TuSC + Spc72N or only Ɣ-TuSC. The concentration of proteins is described in Materials and methods section: “For Figure 2E and Figure 2—figure supplement 2D, purified γ-TuSC was mixed with GST-Spc72N, GST-Spc72N-6His-Stu2^WT^ or GST-Spc72N-6HisStu2^TOGAA^ to a final concentration of 4.6 µM for γ-TuSC and 2.3 µM for the other proteins in HB100 buffer with 10% glycerol.”

3) Ɣ-TuSC oligomerization appears to be enhanced in the presence of Spc72N-Stu2TOGAA (defective in tubulin binding). This experiment should also be done with Spc72 and wildtype Stu2.

We now have performed this experiment with WT Stu2 (Figure 2E; 3 independent experiments) as requested by reviewer 1. The Stu2^TOGAA^ experiment is shown in Figure 2—figure supplement 2D. In all three experiments Stu2^TOGAA^ behaved as Stu2.

3) The authors should quantify Stu2 fluorescence at the cMT minus end in the spc72Δstu2/Kar9-depletion conditions. Is Stu2 completely absent or are factors beyond Spc72/Kar9 also recruiting Stu2 to the minus end?

We have addressed this point in subsection “Stu2 stabilizes γ-tubulin at cMT ends”:

“Depletion of Kar9 in *spc72^∆Stu2^*cells should further decrease Stu2-yeGFP intensity at the minus end of detached cMTs. However, as outlined below, these measurements were complicated by the instability of cMTs in *spc72^∆Stu2^ KAR9-AID kar1∆15* cells and the observation that of the surviving detached cMTs only half carried Tub4 (Figure 6F and G). The normalized Stu2-yeGFP signal intensity at these rare Tub4-positive cMT minus ends was measured with ~0.9, which is similar to what we determined in *spc72^∆Stu2^*cells but clearly higher than the signal in *KAR9-AID* cells (Figure 6B). This suggests that Kar9-AID was not fully degraded in these *spc72^∆Stu2^ KAR9-AID* cells with detached cMTs. Indeed, variations in the efficiency of the auxin-induced degradation of proteins between yeast cells have been reported (Papagiannakis, de Jonge et al., 2017). “

4) As the authors currently present it, the mechanism of Stu2-dependent MT assembly is unclear. The experiments in Figures 6 and 7 are difficult to interpret because controls are lacking. For example, the Stu2 mutants alone were not analyzed in Figure 6. Is there any affect of Stu2 or the Stu2 mutants in the presence of Ɣ-TuSC, but in the absence of Spc72 (Figure 7)?

We have now analysed the Stu2 mutants alone (Figure 3—figure supplement 1). In addition, besides aster formation ability, we have quantified MT formation ability as was done by Roostalu and Surrey in Nat Cell Biol. In response to reviewer 1’s suggestion, we also combined Ɣ-TuSC with Stu2 (Figure 3—figure supplement 1A-CFigure; Ɣ-TuSC+Stu2). In this combination, we see the activity of Stu2. Moreover, we show that Stu2 has to be in the preformed Spc72N-Stu2 complex since addition of individual Spc72N and Stu2 into the reaction mix (which dilutes the proteins and disfavours complex formation) did not show the activity of the Spc72N-Stu2 complex.

Thus, we hope that the data clarify the mechanisms of Stu2 in cMT nucleation. We intensively discuss the function of Stu2 in cMT in the Discussion section.

5) The concentrations of tubulin, and all other proteins used in Figures 6 and 7 are not listed in the methods or legends. This is critical to include. Also, were different concentrations tested? How were the concentrations that were used chosen?

The concentration of tubulin was described in the Materials and methods section (not the legends – I am sorry for this). The final concentration was 12 µM. This concentration was chosen because the Spc110-Ɣ-TuSC rings are able to nucleate MTs at this tubulin concentration. Our tubulin concentration is similar to what Roostalu and Surrey have used in NatCell Biol.

To clarify this point, we write in subsection “Purified Stu2 in complex with Spc72 assembles MT asters”:

“The α/β-tubulin concentration of 12 µM was chosen because it enables the oligomerized Spc110N-γ-TuSC to nucleate microtubules (Kollman et al., 2010, Lin et al., 2014). A similar α/β-tubulin concentration has been used in MT nucleation assays with purified TPX2 and chTOG (Roostalu et al., 2015). Assembled MTs were spun onto cover slips and analysed by fluorescence microscopy. “

6) It is unclear what distinct roles the authors propose for the MT-binding and TOG-binding domains of Stu2 to have in enhanced MT nucleation at the centrosome/ SPB. Are cells with Stu2 MT-binding and TOGAA mutations viable?

This is a good point and we should have added this already in the first submission. Now we show in Figure 3—figure supplement 3F the growth test of the different *STU2* mutants. Moreover, we have heavily modified the Discussion section addressing the role of the TOG domains in Stu2. I agree that the title of the first manuscript version was misleading. TOG domains of Stu2 have an important role in cMT nucleation as indicated by our experiments and outlined in the text.

For example, in the Discussion section we write:

“The TOG domains in the Kar9-Stu2 layer are most likely positioned more distantly away from the cMT minus end than those within Spc72-Stu2 (Figure 8). The two Stu2 layers probably function in a sequential order: Spc72-Stu2 for the addition of the first α/β-tubulin subunits followed by Kar9-Stu2 that adds the next α/β-tubulin molecules. A single XMAP215 molecule bound via its MT binding site to γ-tubulin has with five TOG domains a greater reach to add α/β-tubulin subunits to the growing MT than a single Stu2 layer with only two TOG domains (Widlund et al., 2011). Binding of Stu2 to Spc72 and Kar91 probably compensates for the smaller number of TOG domains in Stu2.”

7) The authors reference two "nucleation activities" of Stu2, but then posit that the TOG domains may be required for extending Ɣ-TuSC-nucleated seeds. This seems contradictory.

As outlined above, we have modified the Discussion section and now better explain the role of Stu2 in MT nucleation.

8) The manuscript could benefit greatly from additional editing for clarity and conciseness. There are many grammatical errors as well. The Figure Legends need to be improved for clarity and completeness.

Thank you very much for this suggestion. I hope that we have done a better job this time.

Reviewer #2:

*[…] The manuscript adds to our understanding of Stu2 and its role in yeast mitosis using a nice blend of* in vitro *and* in vivo *analyses. In its current state it lacks some critical experiments that, if present, would best unite disparate pieces in the paper, and help the authors arrive at some of their conclusions.*

The reviewer states: “The manuscript adds to our understanding of Stu2 and its role in yeast mitosis using a nice blend of in vitro and in vivo analyses.” He/she also points out that the paper lacks critical experiments that should unite different experimental parts.

We have addressed this point of reviewer 2 by re-arrange the manuscript and adding critical experiments for example the new Figure 7.

1) Gel filtration is used to determine whether Spc72 (plus or minus Stu2) causes oligomerization of γ-TuSC, as does Spc110. The results show that Spc72 does shift γ-TuSC to the void volume, and that Stu2 enhances this shift. However, Stu2 also causes a new peak to arise at fraction A11 that could be accounted for by Stu2-mediated dimerization of a γ-TuSC-Spc72 complex. This then calls into question the increase of the γ-TuSC fraction in the void volume: is this really oligomerization as seen with Spc110 which generated γ-TuSC filaments, or is it due to cross-linking mediated by dimers of Spc72 and dimers of Stu2? To resolve this, the authors should determine if the Spc72-Stu2 complex does drive γ-TuSC oligomerization – this could be achieved using negative stain EM analysis, similar to work done by Kollman et al., 2010, DIC microscopy, or fluorescence microscopy.

We have performed the negative staining EM as suggest by reviewer 2. These data are in Figure 2—figure supplement 2E. They clearly show that Spc72-Stu2 oligomerizes Spc72Stu2 into rings that are different in structure compared to the Spc72-Ɣ-TuSC rings. Additional high-resolution EM analysis is needed to understand the structural differences between the Spc72-Stu2-Ɣ-TuSC and Spc72-Ɣ-TuSC rings. This will be done in a future study using cryo-EM.

*2) Figure 7, the authors should conduct a Stu2 WT +/- γ-TuSC control to best determine how much Spc72-Stu2 augments the MT nucleation activity* in vitro.

We have extended the analysis that was shown in Figure 7 (now Figure 3 and Figure 4) by adding the controls that have been asked for by reviewers 1 and 2. The Stu2 WT +/- γ-TuSC control is now shown in Figure 3—figure supplement 3A/B.

*3) The authors delineated independent roles for Spc72 and Kar9 on Stu2 recruitment to the MT minus end, analyzed the effects of these respective complexes on cMT nucleation, number, and length in cells. However, when the authors then move to* in vitro *analyses, they concentrated exclusively on the Spc72-Stu2 interaction (plus and minus γ-TuSC) and its effects on microtubule nucleation* in vitro*. To be comprehensive, it would greatly strengthen the paper to conduct a comparative analysis of the effects of Kar9 and Kar9-Stu2 (plus and minus γ-TuSC) on MT nucleation* in vitro *with that of Spc72-Stu2.*

We appreciate this suggestion. Adding Kar9 in addition to Stu2, Spc72 and γ-TuSC bears the challenge of purifying Kar9 (as Kar9 or Kar9-Bim1 complex – ?) in a native functional state and find buffer conditions that are compatible with all components. This was already challenging enough for the three-component system. I guess, reviewer 2 is suggesting this because we did not explain too well the function of the second Kar9-Stu2 layer in comparison to Spc72-Stu2. We have done this for the new submission in the Discussion section:

“The situation in cells is even more complex since two proteins, Spc72 and Kar9, recruit Stu2 to the minus end of cMTs. The recruitment role of Kar9 is consistent with the reported interactions of Kar9 with Stu2 and γ-tubulin Tub4 (Chen et al., 1998, Cuschieri et al., 2006, Moore and Miller, 2007, Usui et al., 2003) (illustrated in Figure 8).”

“The TOG domains in the Kar9-Stu2 layer are most likely positioned more distantly away from the cMT minus end than those within Spc72-Stu2 (Figure 8). The two Stu2 layers probably function in a sequential order: Spc72-Stu2 for the addition of the first α/β-tubulin subunits followed by Kar9-Stu2 that adds the next α/β-tubulin molecules. A single XMAP215 molecule bound via its MT binding site to γ-tubulin has with five TOG domains a greater reach to add α/β-tubulin subunits to the growing MT than a single Stu2 layer with only two TOG domains (Widlund et al., 2011). Binding of Stu2 to Spc72 and Kar91 probably compensates for the smaller number of TOG domains in Stu2.”

Reviewer #3:

Review of Gunzelmann et al., TOG domain independent microtubule nucleation activity.This paper describes a novel nucleation activity within the spindle pole component, Spc72 that recruits Stu2 (TOG domain protein) to the cytoplasmic face of the spindle pole body, where it increases MT density and plays a role in nucleating MT asters.The binding data in Figure 1 and Figure 2 was very convincing. The data in Figure 3 were problematic. Firstly, they state that incubation in a kar1 mutant results in detached cMTS (Pereira et al., 1993; Erlemann, 2012). I looked at both of these papers and was unable to find the percent of detached vs attached cytoplasmic microtubules in the mutant. It is important to quantitate what fraction of the microtubules are detached to differentiate pole binding from minus-end binding. An alternative interpretation for Figure 3c is that observation of Stu2 in the absence of the stu2 binding site in Spc72 is that these microtubules are not detached from the pole.

40% of cMTs detach from the SPB in *kar1∆15* cells as outlined in Figure 5A. We were quite careful in identifying detached cMTs. We followed the entire cMT to ensure that the cMT was not attached to the SPB.

In cases where we used Tub4-mCherry as marker, the Tub4 signal at the SPB was strong; Tub4 at the cMT minus end was weaker (see Figure 5C). Stu2-yeGFP decorated the wall of the cMT. Thus, we are quite confident that we have only analysed minus ends at detached cMTs.

To make this point clearer, we have now added cartoons to the images (for example Figure 5B and C) illustrating for each case the position of the detached cMT.

The FRAP recovery are nicely done and convincing that there are multiple binding sites.

Thank you.

It is interesting that in this and panel b the mt minus ends seem to be in the shmoo tip. Since there are known plus-end binding complexes in the shmoo tip, is there concern that the co-localization reflects the juxtaposition of the minus-end to the shmoo tip. This is easily addressed by quantitating the fraction of spc72 or stu2 spots in the tip vs. the presence or absence of mt minus ends.

We have done this analysis. The minus end of detached cMTs of *SPC72* and *spc72^∆Stu2^* cells (n=57) frequently localized (~70%) within the schmoo tip periphery (2 µm radius). Detached cMTs of *SPC72* and *spc72^∆Stu2^* cells that lacked Stu2 at the minus end showed a similar schmoo tip preference (11 out of 16 cMT). This orientation is probably a reflection of the association of the minus end motor Kar3 with the schmoo tip {Maddox, 2003}. Kar3 may pull detached cMT minus ends to this localization. In response to this comment of reviewer 3, we now point out in subsection “Spc72 and Kar9 target Stu2 to the cMT minus end” that the cMT minus end frequently localizes within the schmoo tip periphery.

“This cap is the Spc72-γ-TuSC complex (Erlemann et al., 2012) that in ~70% of cells localized within the schmoo tip region (2 µm radius). This orientation is probably a reflection of the association of the minus end motor Kar3 with the schmoo tip (Maddox et al., 2003) pulling the detached cMT minus ends into the schmoo.”

Figure 4D is a critical panel for interpretation of aforementioned experiments. It is stated that a-g are kar1Δ15. Approximately 50% of the cMT are attached in these mutants. Thus, the reviewer needs to redraw Figure 3a to reflect the fact that 50% are detached and 50% are released.

We have modified Figure 5A (former Figure 3A) as suggested by reviewer 3. 40% of *kar1∆15* cells have detached cMTs; 60% do not have detached cMTs. This is now indicated in the Figure 5A.

In addition, the reviewer needs to include the attachment/detachment state of the microtubules for the co-localization. Are they only examining detached microtubules, if so, what is the metric for the data in Figure 3?

In Figure 5 and Figure 6A-C and G we only have analysed detached cMTs because we specifically were interested in the minus end of these cMTs. This analysis is not possible with cMTs that are attached to the SPB. A SPB signal may come from e.g. the binding of Stu2 to a SPB component or its association with the cMT end. In the legend to Figure 5 we state: “(b-f) Experiments were performed with *kar1Δ15* cells treated with α-factor and analysis was done on single, detached cMTs.” Similarly, we state in the legend to Figure 6A-C and g that detached cMTs were measured.

In Figure 6D and E we have included all cMT categories: (i) SPB attached cMTs, (ii) detached cMTs, and (iii) no cMTs. This analysis gives us an overview how the different populations behave in mutant cells and over time.

The metric in Figure 4 is clear, mCherry-Tub1, but if 50% are attached, the data without Tub1 become significantly ambiguous.

I do not fully understand the concern of reviewer 3. Figure 4 (now Figure 6) shows mostly cells with mCherry-Tub1 signal (Figure 6C and D-G). Figure 6A shows co-localization of Kar9 and Spc72 (minus end marker) at the detached cMT minus end. Kar9 decorates in part the detached cMT. Figure 6—figure supplement 4B confirms Kar9-GFP at a detached cMT labelled with mCherry-Tub1. The quantification in Figure 6B was done in *TUB4-mCherry STU2-GFP* cells. The Tub4-mCherry signal gives the position of Stu2 at the cMT minus end. Stu2 also decorates the detached cMT (see Figure 5B for two examples). We only analysed cells where we unambiguously could identify the nuclear MT bundle and the single detached cMT to be sure that only the minus end of detached cMTs was analysed.

In summary, this paper documents an important second TOG-independent mechanism in Stu2 for MT formation. Stu2 has critical roles in spindle orientation and kinetochore function, and thus this second site is important to understand. The authors should ensure that for all reports of minus-end binding, that these are indeed detached cytoplasmic microtubules.

Thank you.

[Editors’ note: the author responses to the re-review follow.]

[…] The study expands our understanding of MT nucleation in yeast. It adds to the field in a unique way and complements the recently published XMAP215/γ-tubulin work from the Petry lab.However, before the paper can be published, we would like to ask you to address some remaining issues which concern data analysis, paper organization and writing.Data analysis and interpretation:1) The authors state that the aster density is lower in Spc72N-Stu2TOGAAA asters than in Spc72N-Stu2 asters. However, the quantification used to determine aster density is 'MT formation activity', or the mean intensity fluorescence of the entire field. That type of measurement encompasses the entirety of MT assembly. Therefore, a better quantification of aster density is required.

We now included an additional quantification of aster fluorescence intensity, which reflects the aster density (See new Figure 3F).

Also, it appears that the example images shown in Figure 3E are saturated, making aster density difficult to assess.

All image settings were chosen to be exactly the same and they are optimised to visualise single MTs. This is why some images in the centre are saturated. However, for quantification (new Figure 3F) we only analysed images in the linear range without saturation.

2) Is there a suspected reason for the increase in Stu2 fluorescence in spc72Stu2Δ versus Spc72 WT cells (Figure 5E)? If you remove the CM1 region of Spc72 in yeast, is there a decrease in Stu2 fluorescence at the cMT minus end?

The suggested experiment using the *Spc72^ΔCM1^* is complicated to realise. As we have shown in Figure 1B this mutation is lethal. Instead, we tested the possibility that enhanced recruitment of Kar9 could be responsible for the increased recruitment of Stu2 in *spc72^Δstu2^* mutant cells. Indeed, it turned out that there is a higher availability of Kar9 at the minus end of detached cMTs in *Spc72^Δstu2^* cells compared to *SPC72* cells as it is now shown in the new Figure 6—figure supplement 1C.

3) The authors state that the Stu2 signal in spc72Stu2Δ KAR9-AID +IAA cells is similar to spc72Stu2Δ. The value stated (~0.9) is actually even more similar to the Stu2 fluorescence in Spc72 WT cells (Figure 5E). This quantification, even if it is for a small number of cMTs, should be shown in the figure.

We now included the quantification as Figure 5E. However, as the number of quantified cells is so low we did not performed a statistic analysis.

Additionally, it is suggested that the KAR9-AID was not fully degraded under these conditions, and that is why Stu2 is still recruited to the cMT minus end. However, a complete knockdown of KAR9-AID in another yeast strain (Figure 6—figure supplement 1C) is shown. The authors should demonstrate the extent of KAR9-AID degradation in spc72Stu2Δ KAR9-AID +IAA cells. Alternatively, the authors should fluorescently tag the KAR9-AID protein to determine if the few numbers of cMTs that still maintain Stu2 also have Kar9 at the cMT minus ends. A possible interpretation is that there may be an additional factor recruiting Stu2 to cMT minus ends.

Depletion of a human centrosomal protein by siRNA appears very efficient based on an immunoblot analysis of cell extracts. However, in IF quite often a residual signal is observed at the centrosome. Γ-Tubulin siRNA is a good example for such behaviour. According to our analysis about 5% of cells carry some residual amount of Kar9-AID at the minus end of cMTs. We do not know how much Kar9 is needed for the recruitment of functional levels of Stu2. In any case, Kar9-AID levels are reduced by at least 95% compared to *KAR9* cells, probably even more. This reduction will be below the detection limit of the immunoblot. To address this point of the reviewers, we added in subsection “Stu2 stabilizes γ-tubulin at cMT ends” following sentence.

“This residual amount of Kar9-AID in the small number of *spc72^∆Stu2^ KAR9-AID* cells with cMTs was, however, below the detection limit of the immunoblot in the new Figure 6—figure supplement 1D.”

4) Comments on the FRAP analysis: The authors claim that there are two pools, a stable and unstable fraction of Stu2 at the minus-ends of microtubules. There is concern that there is a limited pool of soluble Stu2, and the recovery curves (Figure 5F) potentially misleading. The lack of recovery in one sample could be due to limited protein pool (so there is not enough unlabeled protein around to populate the FRAP zone. The authors could report the average fluorescence intensity of the spots (in arbritrary units) to examine whether spots of the same fluorescence are being targeted in each sample.

To rule out the possibility that a limitation of freely available cytoplasmic Stu2-yeGFP is the reason for incomplete recovery, we quantified the fluorescence signal intensity of the examples shown in Figure 5F before bleaching. It should be noted, that the examples shown and that have been used for the analysis have a higher intensity in Stu2-yeGFP for *Spc72^Δstu2^* compared to *SPC72* cells in the beginning of the experiment.

Improving the organization and writing of the paper:1) The authors have a lot of interesting data, but because of the complex nature of Stu2's multiple roles as well as the complexity of some of the experiments, a lot of the intricacies are lost upon first reading. It would be specifically helpful to have a section in the introduction that clearly outlines each putative role and interaction of Stu2 that will be tested, and the general strategy used to test each of these roles/interactions.

We are grateful for the suggestion of the reviewer to improve the overall comprehensibility of the paper. We now included a section at the beginning of each result paragraph that refines the question that we wanted to address and the strategy how to test for possible roles/interactions.

In the Introduction we write: “However, there are still the following open questions: (1) Do TOG domain proteins have the ability to nucleate MTs in vivo independently of γ-tubulin complexes? (2) Do TOG domain proteins have structural functions at the MT minus end? (3) Is the role of TOG domain proteins in MT nucleation context specific (Roostalu and Surrey, 2017)?“

2) Before each set of experiments, the authors should clearly explain why they are performing the experiment and what they expect to occur (a hypothesis/model). In many areas this information is lacking or vague. For example, subsection “Stu2 stabilizes γ-tubulin at cMT ends” "To further elevate the role of Stu2 at the cMT minus end…". It's unclear what the authors mean by that and it is difficult to follow their line of logic. As much as possible, the authors should reference back to their original models so that the reader can more easily follow.

This goes in line with the first point and as suggested we tried to improve the understandability by rephrasing/rewriting the mentioned paragraph and the results section in general.

3) It appears that the authors fit their MST data to a single-site model, but this should be clearly stated in the Materials and methods section.

Indeed, we used a single-site model for fitting our MST data and we now sate this in the Materials and methods section.

4) The grammar, primarily in the Introduction, needs some additional work.

We tried to improve this point as suggested and modified the Introduction.

5) The authors frequently use the uncommon term: "in dependence of" referring to an effect that is "dependent". Given the wide international readership of eLife, we suggest that this phrase be rewritten in a more active form so that people, who do not speak English as a first language, do not confuse the statement with "independent of.."

We changed this term to avoid confusion, as implied.

6) Subsection “Purified Stu2 in complex with Spc72 assembles MT aster” fifth paragraph is confusing and should be rewritten.

We rephrased this paragraph in order to make it clearer.

7) Discussion section, ninth paragraph, the authors probably mean "ratios" rather than "rations".

We appreciate the notion of this typo and changed it accordingly.

8) Figure 7—figure supplement 1D and E are important and should probably be included in a main figure.

We changed the appearance of the Figure as suggested by the reviewer and moved the Figure 7—figure supplement 1D and E to Figure 7E and F.

9) Figure 8 needs a figure legend and the objects within the figure need to be clearly labeled.

We added a figure legend and all components shown in this Figure are now labelled.

10) Figure 7A, B need labels indicating what is fluorescently tagged.

We have added the missing label.

11) Several times protein constructs or yeast strain acronyms aren't described upon first mention. Upon the first introduction of a new protein construct that is given an abbreviation, a description and the amino acid modifications (if applicable) should be stated.

We carefully checked the entire manuscript and introduced missing descriptions or abbreviation at the first time of appearance.